

# Retrieval of Lower-Order Moments of the Drop Size Distribution using CSU-CHILL X-band Polarimetric Radar: A Case Study

Viswanathan Bringi[1], Kumar Vijay Mishra[2], Merhala Thurai[1], Patrick C. Kennedy[3], and Timothy H. Raupach[4]

[1]Department of Electrical and Computer Engineering, Colorado State University, Fort Collins, Colorado, USA
[2]United States Army Research Laboratory, Adelphi, Maryland, USA
[3]Retired, Colorado State University, Fort Collins, Colorado, USA
[4]Department of Geography and Oeschger Centre for Climate Change Research, University of Bern, Bern Switzerland. Present address: Climate Change Research Centre, University of New South Wales, Sydney, Australia

**Correspondence:** V. N. Bringi (bringi@colostate.edu)

**Abstract.** The lower order moments of the drop size distribution (DSD) have generally been considered as difficult to retrieve accurately from polarimetric radar data because these data are related to higher order moments. For example, the $4.6^{\text{th}}$ moment is associated with specific differential phase, $6^{\text{th}}$ moment with reflectivity and ratio of high order moments with differential reflectivity. Thus, conventionally, the emphasis has been to estimate rain rate ($3.67^{\text{th}}$ moment) or parameters of the exponential or

gamma distribution for the DSD. Many double-moment "bulk" microphysical schemes predict the total number concentration (the $0^{\text{th}}$ moment of $M_0$) and the mixing ratio (or equivalently, the $3^{\text{rd}}$ moment $M_3$). Thus, it is difficult to compare the model outputs directly with polarimetric radar observations or, given the model outputs, forward model the radar observables. This article describes the use of double-moment normalization of DSDs and the resulting stable intrinsic shape that can be fitted by the generalized gamma (G-G) distribution. The two reference moments are $M_3$ and $M_6$ which are shown to be retrievable using

the X-band radar reflectivity, differential reflectivity and specific attenuation (from the iterative ZPHI method). Along with the climatological shape parameters of the G-G fit to the scaled/normalized DSDs, the lower order moments are then retrieved more accurately than possible hitherto. The importance of measuring the complete DSD from 0.1 mm onwards is emphasized using, in our case, an optical array probe with 50 $\mu$m resolution collocated with a two-dimensional video disdrometer with about 170 $\mu$m resolution. This avoids small drop truncation and hence the accurate calculation of lower order moments. A case study of a

complex multi-cell storm which traversed an instrumented site near the CSU-CHILL radar is described for which the moments were retrieved from radar and compared with directly computed moments from the complete spectrum measurements using the aforementioned two disdrometers. Our detailed validation analysis of the radar-retrieved moments showed relative bias of the moments $M_0$ through $M_2$ was $< 15\%$ in magnitude, with Pearson's correlation coefficient $> 0.9$. Both radar measurement and parameterization errors were estimated rigorously. We show that the temporal variation of the radar-retrieved characteristic

diameter with $M_0$ resulted in coherent "time tracks" that can potentially lead to studies of precipitation evolution that have not been possible so far.



## 1 Introduction

The principal application of polarimetric radar has historically been directed towards more accurate estimation of rain rate ($R$) that is driven largely by the operational agencies for hydrological applications. It now strongly appears that, as a major

step forward, the operational algorithm for the US Weather Surveillance Radars - 1988 Doppler (WSR-88D) network will be based on specific attenuation because, among other advantages, it is linearly related to rain rate at S-band (*Ryzhkov et al.* 2014; *Cocks et al.* 2019; *Wang et al.* 2019). This method has also been evaluated quite extensively at X-band by *Diederich et al.* (2015), where the specific attenuation ($A_h$) is much larger than at S-band but not linear with $R$. The development of $R(A_h)$ algorithms rests on a large body of work since the early 1990s and is related to attenuation-correction using differential

propagation phase as a constraint (*Bringi et al.* 1990; *Smyth and Illingworth* 1998; *Testud et al.* 2000; *Bringi and Chandrasekar* 2001; and references therein).

The retrieval of drop size distribution (DSD) parameters has also been a strong impetus for radar polarimetry. In this context, there exists a large body of literature that is based mainly on the unnormalized (*Ulbrich* 1983) or normalized (*Illingworth and Blackman* 2002; *Testud et al.* 2001) gamma model. This model is parameterized by a set of three quantities, namely $\{N_0, \mu, \Lambda\}$

or $\{N_w, \mu, D_m\}$, where $N_0$ and $N_w$ are "intercept" parameters, $\mu$ is the shape factor, $\Lambda$ is the "slope", $D_m$ is the ratio of the $4^{\text{th}}$ to $3^{\text{rd}}$ moments of the DSD $N(D)$, and $D$ is the diameter of the raindrop (*Ryzhkov and Zrnić* 2019; and references therein). The gamma model has also been used in the double-moment "bulk" microphysical schemes that predict the mass mixing ratio (or, equivalently the third moment $M_3$; for moment order $k$, we write $M_k$ and the total concentration of drops (or, $M_0$) (e.g., *Meyers et al.* 1997). The lower order moments of the DSD ($M_0$ through $M_{3+b}$, where $b$ is the exponent of

the fall-speed-D power law), are important in describing various microphysical processes such as collisional (break-up and coalescence), evaporation, sedimentation (e.g., *Milbrandt and Yau* 2005). However, radar polarimetry has not been focused on these lower order moment retrievals because the radar observables (horizontal) reflectivity $Z_h$, differential reflectivity $Z_{dr}$, and specific differential phase $K_{dp}$ are related to the higher order moments such as $M_6$, the ratio $M_7/M_6$ and $M_{4.5}$, respectively.

Defining a scaled diameter $x = D/D_m$, the *normalized DSD* is a function of $x$ as $h(x) = N(D)/N_w$. The observation of

45 *Testud et al.* (2001) regarding the "remarkable" stability of the shape of $h(x)$ using measured DSDs was a significant advance because they did not impose an *a priori* form for $h(x)$. Apart from the shape "stability" of h(x), they also showed a large compression in the "scatter" of $h(x)$ compared to N(D). While they did not refer to their normalization as double-moment using $M_3$ and $M_4$ as the reference moments, *Lee et al.* (2004; henceforth, *L04*) generalized the scaling/normalization framework by introducing any two reference moments $M_i$ and $M_j$ of any order $i, j > 0$. As per this framework, in a compact notation,

$N(D) = N_0' h(x)$ with a different $x = D/D_m'$, where $N_0' = M_i^{(j+1)/(j-i)} M_j^{(i+1)/(i-j)}$ and $D_m'$ is the ratio of $(M_j/M_i)^{1/(j-i)}$. In essence, the variance of the DSDs due to different rain types and intensities is largely controlled by the variability in $N_0'$ and $D_m'$ and much less so by $h(x)$. Further, any moment $M_k$ can be expressed as power laws of $M_i$, $M_j$, and the $k$-th moment of $h(x)$. *L04* also recognized that if $h(x)$ is assumed to follow the generalized gamma (G-G) model with two shape parameters, then it could fit most naturally occurring DSD shapes. We refer the reader to *Stacy* (1962) for the expressions of the probability

density function (pdf) of the G-G and its moments. The G-G form has been applied to model cloud droplet spectra, ice crystal





and snow spectra (*Delanoë et al.* 2014) as well as raindrop spectra (*Raupach and Berne* 2017a; *Thurai and Bringi* 2018). The three moment normalization for this model is provided in *Szyrmer et al.* (2005). The generalization to N-moment normalization scheme given in *Morrison et al.* (2019) does not specify any particular form for $h(x)$ other than that its moments should be finite. In agreement with *Szyrmer et al.* (2005), they found that three-moment normalization was sufficient to "compress" the

scatter of $h(x)$ (they term it as $g(x^*)$). Further, it minimized the errors in the estimation of the other moments expressed as power laws of the reference moments. For remote sensing applications (cloud and drizzle), they found that the set of moments $\{M_2, M_3, M_6\}$ was one possible choice mentioning lidar backscatter ($M_2$), microwave attenuation ($M_3$) and radar reflectivity ($M_6$). While the combination of $M_3$ and $M_6$ was not optimal for estimating the lower order moments (in particular, $M_0$), it was better than using $M_6$ alone.

Recently, using the double-moment approach of *L04*, *Raupach and Berne* (2017a; *RBa*) showed that measured DSDs in stratiform rain with $h(x)$ expressed in the G-G form have shape factors that are sufficiently "invariant" for practical use across different rain climatologies if the reference moments are chosen carefully. Their result essentially validated the "remarkable" stability conclusion of $h(x)$ by *Testud et al.* (2001) which was based on limited data in oceanic rain. *RBa* speculated that the transition (i.e., between convective and statiform rain) and convective rain DSDs would also have a sufficiently "invariant"

$h(x)$ but they did not have a large enough database to make such a conclusion.

The polarimetric (X-band) radar-retrieval of moments using reference moments $M_3$, $M_6$ and an "invariant" $h(x)$ of the G-G form were first described in *Raupach and Berne* (2017b; *RBb*). Their retrieval of $M_6$ was based on $Z_h$ while $M_3$ was retrieved in a two-step procedure using $Z_{dr}$ and $K_{dp}$. We discuss their results in detail later in this paper. Here, it suffices to mention that their measured DSDs were based on a network of Parsivel disdrometers, which did not have the resolution to measure the

shape of $h(x)$ for $D < 0.75$ mm or so (as shown later by *Raupach et al.* 2019). Additionally, with "noisy" $Z_{dr}$ and $K_{dp}$ radar data (for $Z_h < 37$ dBZ), their validation of the moments ($M_0$ through $M_7$) using radar measurements was not conclusive but sufficient to demonstrate that their approach gave results similar to other methods based on normalized gamma model using "more classical" radar-retrievals of $\{N_w, \mu, D_m\}$ (*Gorgucci et al.* 2008; *Kalogiros et al.* 2012).

Whereas *RBb* used measured DSDs and the polarimetric radar forward operator to derive the retrieval algorithms for $M_3$

and $M_6$, there has been a reverse moment-based polarimetric forward operator (*Kumjian et al.* 2019). This reverse approach employs a very large database of measured and bin-resolved one-dimensional (1-D) model output DSDs to build a look-up table that maps the various moment pairs to the expected values of $Z_h$, $Z_{dr}$, and $K_{dp}$ along with their standard deviations. Their application was to determine the moment pairs that could potentially be prognosed in numerical microphysical schemes of rain, would be "optimally" constrained by polarimetric radar measurements. They found that the pair $\{M_6, M_9\}$ was optimal

in terms of lowest variability in $\{Z_h, Z_{dr}, K_{dp}\}$ but that the pair $\{M_3, M_6\}$ was suboptimal but still "useable". Thus, *RBb*'s choice of $\{M_3, M_6\}$ as the two reference moments was validated by *Kumjian et al.* (2019). The rationale through which $M_9$ (whose sampling error would be very large using available disdrometers) entered the moment pair is not entirely clear. It could be because of correlating $Z_{dr}$ with absolute moments ($M_0$ through $M_9$) as opposed to the more physically-based ratio of moments such as $D'_m = (M_6/M_3)^{1/3}$ in *RBb* or $M_7/M_6$ as in *Jameson* (1983).



This work further develops on *RBb* but, instead of $K_{dp}$, we employ specific attenuation ($A_h$) given its operational use in estimating $R$. The iterative ZPHI algorithm, which is a variant of *Testud et al.* (2000), is used here to estimate $A_h$ (*Bringi et al.* 2001; *Park et al.* 2005a,b). The reference moment $M_3$, which is proportional to rain water content ($W$) is retrieved using a modification of *Jameson* (1993) by fitting $A_h/W$ as a smoothed cubic spline with $D_m$. The prior step is the retrieval of $D'_m$ from $Z_{dr}$ and then retrieving $D_m$ from $D'_m$. This multi-step procedure minimizes the parameterization errors. As in *RBb*, the

reference moment $M_6$ is derived as power law fit to $Z_h$. The other major difference with *RBb* is the use of collocated optical array probe (50 $\mu$m resolution) and two-dimensional video disdrometer (2DVD) inside a double-fence wind shield (*Thurai et al.* 2017a). The "complete" DSD was measured from 0.1 mm onwards thus avoiding truncation at the small drop end. This leads to more accurate estimates of the lower order moments as well as more accurate $h(x)$. The methodology of G-G fits to $h(x)$ are described in *Thurai and Bringi* (2018) and *Raupach et al.* (2019). The use of a very narrow (0.33°) beam at X-band with high gain and a short vertical distance from radar pixel to the instrumented site were additional factors that differed from

*RBb*. We also show coherent "time tracks" in the $D_m$ versus $M_0$, $D_m$ versus $W$, and $D'_m$ versus $M_6$ planes, where all variables are based on radar retrievals.

The rest of the article is organized as follows. In the next section, we briefly discuss the surface instrumentation (disdrometers). The CSU-CHILL radar and its use in characterizing the multi-cell storm complex (the case study) as well as data extracted

over the instrumented site are given in Section 3. The retrieval of the two reference moments $\{M_3, M_6\}$ follows in Section 4. Several different ways of validating the moment retrievals are presented in Section 5. We follow this by a discussion in Section 6 and summarize the case study in Section 7. We cannot draw firm conclusions from just one case study even though the analysis is quite detailed. Rather this work may be considered as a proof-of-concept that will require further validation to be undertaken in the future. It is difficult to find radar data with revisit times < 90 s over an instrumented site unless a dedicated

experiment is proposed and funded. In our case, the event of 23 May 2015 was largely a target of opportunity as one of the co-authors (PCK) had the foresight to collect data on this day without considering that it would lead to a detailed case study of moment retrievals. An Appendix provides procedures for estimating the radar measurement error contribution to the variances of, firstly, the reference moments $\{M_3, M_6\}$ and then the variances of the other moments. The estimates of the variances of the ratio of correlated variables of the form $X^p Y^{-q}$ are derived using a Taylor expansion to $2^{\text{nd}}$ order. The parameterization error

variances are estimated for $\{M_3, M_6\}$ and summed with the radar measurement error variances to yield the total error variance for each moment retrieved.

Throughout this paper, we use "H" as a subscript for reflectivity $Z_H$ to denote units of dBZ at horizontal polarization. The lower case "h" in $Z_h$ means units of mm$^6$ m$^{-3}$. The same applies to $Z_{DR}$ (in dB) or $Z_{dr}$ (ratio). The functions $\text{Var}(\cdot)$ and $\overline{(\cdot)}$ yield the variance and mean of their arguments, respectively. We use $\text{Cov}(X, Y)$ for the covariance between the variables $X$

and $Y$. A set is denoted by curly brackets $\{\cdot, \cdot, \cdot\}$. The notation $\mathbb{E}\{\cdot\}$ is used for the statistical expectation; $\langle\cdot\rangle$ for the average of its argument; $\Gamma(\cdot)$ for the gamma function; and $\mathbb{Im}\{\cdot\}$ for the imaginary part of its complex argument.



## 2  Surface Instrumentation

The principal surface-based instruments used in this study are the MPS (or Meteorological Particle Spectrometer, manufactured by Droplet Measurement Technologies) and $3^{rd}$ generation 2DVD, both located within a $2/3$-scale Double Fence Intercomparison Reference (DFIR; *Rasmussen et al.* 2012) wind shield. As reported in *Notaroš et al.* (2016), the $2/3$-scale DFIR was effective in reducing the ambient wind speeds by nearly a factor of 3 based on data from outside and inside the fence. An OTT Pluvio rain gage was also available for rain rate and rain accumulation comparison with the disdrometers.

The instrument set-up was the same for the two sites, i.e., at Greeley, Colorado (GXY) and at Huntsville, Alabama (HSV). The case study of 23 May 2015 occurred near the CHILL radar site in Colorado. As will be described later the radar retrieval algorithms of the reference moments $M_3$ and $M_6$ were based on scattering simulations from DSD data from both sites. Huntsville has a very different climate from Greeley, and its altitude is $212$ m mean sea level (MSL) as compared with $1.4$ km MSL for Greeley. According to the Köppen–Trewartha climate classification system (*Trewartha and Horn* 1980), Greeley has a semiarid-type climate, whereas Huntsville has a humid subtropical-type climate (*Belda et al.* 2014).

The MPS is an optical array probe (OAP) that uses the technique introduced by *Knollenberg* (1970, 1976, 1981) and measures drop diameter in the range from $0.05 - 3.1$ mm (but the upper end of the usable range is limited to $1.5$ mm due to reduced sampling volume). A 64-element photo-diode array is illuminated with a $660$ nm collimated laser beam. Droplets passing through the laser cast a shadow on the array, and the decrease in light intensity on the diodes is monitored with the signal processing electronics. A two-dimensional image is captured by recording the light level of each diode during the period that the array is shadowed. The limitations and uncertainties associated with OAP measurements have been well documented (*Korolev et al.* 1991, 1998; *Baumgardner et al.* 2017). The sizing and fall speed errors primarily depend on the digitization error ($\pm 25$ $\mu$). The fall speed accuracy according to the manufacturer (DMT) is $< 10\%$ for $0.25$ mm and $< 1\%$ for sizes greater than 1 mm, limited primarily by the accuracy in droplet sizing. To calculate $N(D)$, the measured fall speed is not used. Rather a cubic polynomial fit from the manufacturer (DMT) is employed. Details of the calculation of $N(D)$ are given in the Appendix of *Thurai et al.* (2017a) and updated in *Raupach et al.* (2019).

The $3^{rd}$ generation 2DVD is described in detail by (*Bernauer et al.* 2015). Its operational characteristics are similar to earlier generations. In particular, the accuracy of size and fall speed measurement has been well documented (e.g., *Schönhuber et al.* 2007; *Schönhuber et al.* 2008; *Thurai et al.* 2007, 2009; *Huang et al.* 2008). Considering the horizontal pixel resolution of about $170$ $\mu$m and other factors, the effective sizing range is $D > 0.7$ mm (*Thurai et al.* 2017a). The fall velocity accuracy is determined primarily by the accuracy of calibrating the distance between the two orthogonal light "sheets" or planes and is $< 5\%$ for fall velocity $< 10$ m s$^{-1}$ *Schönhuber et al.* (2008). A comparison of fall speeds between the MPS and 2DVD have been reported by *Bringi et al.* (2018) from both the GXY and HSV sites with excellent agreement. The only fall velocity threshold used for the 2DVD is the lower limit set at $0.5$ $ms^{-1}$ in accordance with the manufacturer guidelines for rain measurements. The instrument is designed to prevent drops from entering the housing where the cameras are positioned. Without going into details, it suffices to mention that small drops can enter via slits that allow the light to illuminate the cameras or drops can hang on the slits. Both of these effects cause spurious images that the matching software cannot reject





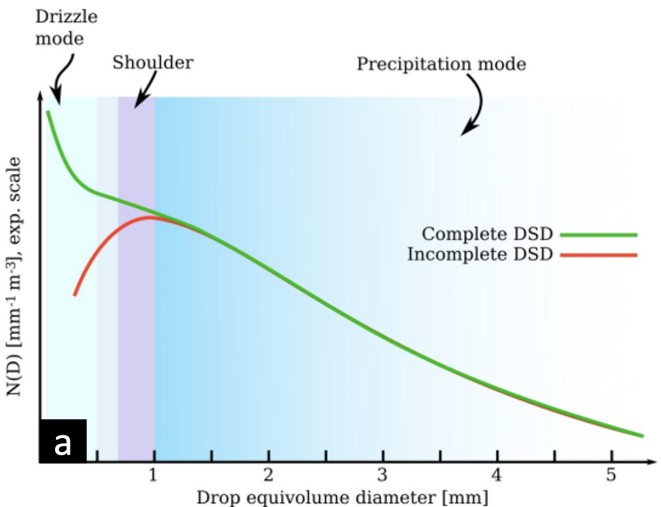

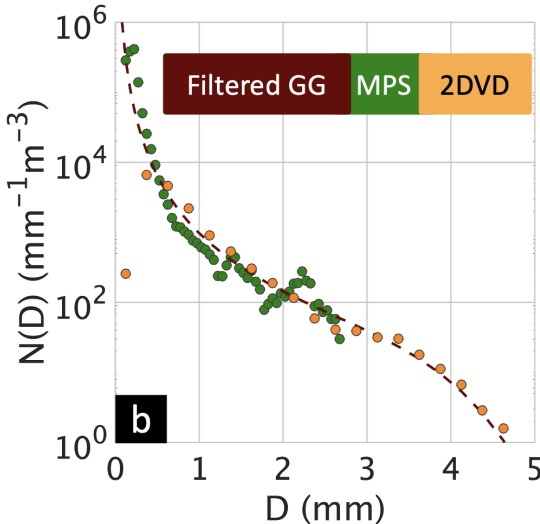

**Figure 1.** (a) Conceptual illustration of the complete DSD comprising the drizzle mode, the "shoulder" region and the precipitation mode. The incomplete DSD is due to drop truncation by instruments that cannot measure the concentration of small drops. (b) An example of measured 3-min averaged DSD ($R \approx 60$ mm h$^{-1}$) using collocated optical array probe with a 2DVD showing the separate measurements (note the high resolution of the MPS and the drop truncation of the 2DVD). The composite or compete spectrum is obtained by using the MPS for $D \leq 0.75$ mm and 2DVD for $D > 0.75$ mm. The dashed line is the G-G fit (with parameters $\mu = -0.3$, $c = 6$; see Eq. (1) for details) to the complete spectrum. Data from 23 May 2015 case study at 2045 UTC.

(*Larsen and Schönhuber* 2018). Thus, caution is necessary when using the 2DVD fall speeds for sizes $< 0.6$ mm (about $3 - 4$ pixels).

In our application, we utilize the MPS for measurement of small drops with $0.1 \leq D < 0.75$ mm and the 2DVD for larger sized drops (see *Raupach et al.* 2019 for the rationale). This is termed here as the "complete" size spectrum and $2,928$ 3-min averaged spectra were available from the two sites with minimum rain rate of $0.1$ mm h$^{-1}$ and maximum of $286$ mm h$^{-1}$. More details of the rainfall types, measurement time periods, comparison with gages and related analyses are available in *Thurai et al.* (2017a) and *Raupach et al.* (2019). Figure 1a illustrates the "complete" DSD with the "drizzle" mode (*Abel and Boutle* 2012), the "shoulder" (which depending on the rain rate) can extend to $\approx 2$ mm, and the precipitation mode. The "incomplete" spectra, in which small drops are not measured accurately due to resolution, sensitivity or other issues (2DVD or Parsivel; *Park et al.* (2017)) frequently shows the convex down shape at the small drop end. Here, we only use the complete $N(D)$ by compositing the data from MPS and 2DVD. An example is shown in Fig. 1b which illustrates the main features of the "complete" DSD during the time that peak rain rate (3-min averaged $R$ of $60$ mm h$^{-1}$) was occurring at the instrumented site during the 23 May 2015 event. The shape is equilibrium-like but with a single "drizzle" mode, a well-defined shoulder and faster (than exponential) fall off in the tail (*Low and List* 1982; *McFarquhar* 2004; *Straub et al.* 2010).



**Table 1.** Technical specifications of CSU-CHILL X-band channel with the dual offset Gregorian antenna

| Parameter | Value |
|---|---|
| Main reflector diameter | 8.5 m |
| Main beam with (3 dB) | $0.33°$ |
| Maximum sidelobe levels | $< -36$ dB |
| Operating frequency | 9.41 GHz |
| Peak transmit power (magnetron) | 25 kW total; split between H and V |
| Sensitivity at 10 km range | -15 dBZ |
| Range gate length | 90 m |

## 3   CSU-CHILL Radar

The CSU-CHILL (Colorado State University-University of Chicago/Illinois State Water Survey) radar is described in *Brunkow et al.* (2000); *Bringi et al.* (2011a). Details on the conversion to a dual-wavelength system and the current radar specifications are given in *Junyent et al.* (2015) (see condensed version in Table 1).

Suffice to state here that the X-band polarimetric mode is "simultaneous transmit and receive" or SHV and the 3-dB beam width is very narrow at $0.33°$ with gain of 55 dB. There are three separate feed or orthogonal mode transducers (OMTs) available: (a) an S-band feed (beam width in far-field is $1°$) whose performance is described in *Bringi et al.* (2011a), (b) an S-X band dual-wavelength feed that was used in the data described herein, and (c) an X-band feed. For the 23 May 2015 event, our retrievals of the reference moments $\{M_3, M_6\}$ are based on the X-band polarimetric data $\{Z_h, Z_{dr}, \Phi_{dp}\}$ only, where $\Phi_{dp}$ is differential phase shift. Only X-band data were used even though S-band data were available simultaneously. Our choice for using the X-band data was due to the very high resolution provided by the $0.33°$ beam and the larger range of values for the X-band specific attenuation for a given rain rate (relative to S-band). The case study convective event was a complex of multiple cells with strong azimuthal and elevation gradients across the "echo cores" which generally precludes accurate dual-wavelength estimation of range-resolved specific attenuation. The narrow X-band beam also allows a lower elevation angle ($1.5°$) to be used before clutter contaminates the signal. The instrumented site was located at Easton which is 13 km SSE of the radar (along the $171.25°$ azimuth). Details of the terrain variation between the radar and the Easton site are given in *Kennedy et al.* (2018).

### 3.1   Brief Description of Storm Characteristics from Radar

The synoptic environment on 23 May 2015 was conducive to thunderstorm development in northeastern Colorado. A low at the 500 hPa level was analyzed over Utah at 12 UTC. This system was forecast to move eastward and promote upward motion within the moist air mass that was in place over the eastern plains of Colorado. In recognition of this situation, the Storm Prediction Center (SPC) Convective Outlook valid for the afternoon hours included a slight risk of severe thunderstorm

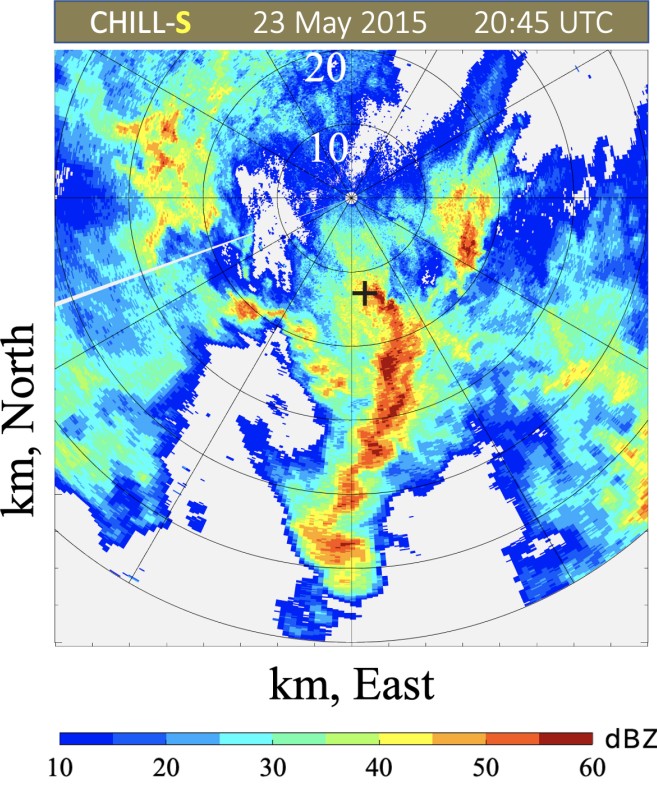

**Figure 2.** Low elevation angle ($1.5°$) PPI sweep of S-band reflectivity ($Z_H$) at 2045 UTC. The "+" marks the location of the instrumented site (MPS and 2DVD).

development over northeastern Colorado. Persistent low cloud coverage ended up limiting surface heating within ~50 km of CSU-CHILL, reducing thunderstorm intensity. The SPC storm reports did not contain any severe category hail (diameter of 2.54 cm or larger), or surface wind speeds of $25 \ \mathrm{m \ s^{-1}}$ or more. Volunteer weather observers reported several instances of small

(0.64 cm or less) hail mostly in the Rocky Mountain foothills ~60 km southwest of the radar. Afternoon surface temperatures were ~14°C in the CSU-CHILL / Greeley area. The 0°C level in the Denver late afternoon sounding was at ~3.5 km MSL (2.1 km above ground level [AGL]).

Figure 2 shows a low elevation angle ($1.5°$) plan position indicator (PPI) scan of S-band $Z_H$ at 2045 UTC which was the time of peak rainfall at the instrumented site (also referred to as Easton) identified by the + marker in Fig. 2. The main echo

feature is the near N-S orientation of multiple 55 dBZ cores extending from the Easton site to nearly 50 km to the south. The rainfall over the site lasted for over 90 mins, and PPI scans at fixed $1.5°$ elevation angle were repeated every 90 s. This good time resolution enabled the validation of the moment retrievals which otherwise would not have been possible (for example, with WSR-88D scan cycle times of around 5 mins).





The general echo movement near Easton was estimated at $10 \, \mathrm{m}s^{-1}$ towards the radar on average from the south. After the peak echo of $55$ dBZ traversed the instrumented site, another cell produced very heavy rain at the radar site with no evidence of graupel/hail (visual observations by one of the co-authors, PCK). One volunteer observer located $15$ km east of the Easton instrumentation site reported $0.64$ cm hail mixed with heavy rain between 2030 and 2045 UTC. The CSU-CHILL radar data showed that this small hail was generated by an isolated, higher reflectivity cell that was separated from the storms that crossed the instrumented site. We do not believe that hail occurred at the Easton site during the analyzed time period, as also confirmed by 2DVD fall speed observations.

There was no RHI scan at 2045 UTC. Therefore, the vertical echo structure could not be determined at this time close to the peak rainfall over Easton, but range-height indicator (RHI) scans about $18$ min earlier performed along the $172°$ azimuth are shown in Fig. 3. The top panel shows measured reflectivity at X-band (uncorrected for attenuation) while the measured S-band reflectivity is plotted in the bottom panel. Several strong cells ($> 55$ dBZ) are noted south of Easton at ranges of 27 and 32 km; the cell at 32 km shows significant attenuation. However, there is no significant attenuation at 13 km range where the instrumented site is located. The 10 dBZ echo top reaches 8 km AGL.

## 3.2 Attenuation Correction

As mentioned earlier, we use the X-band radar for quantitative moment retrievals. It is apparent that the strong cells will cause attenuation so the X-band measured $Z_h$ and $Z_{dr}$ have to be corrected for attenuation and differential attenuation, respectively. The method used herein is exactly the same as described in *Mishra et al.* (2016). For correcting the measured $Z_h$, we apply an iterative version of the ZPHI method, which uses a $\Phi_{dp}$ constraint (*Testud et al.* 2000; *Bringi et al.* 2001) that was originally developed at C-band but later extended to X-band by *Park et al.* (2005a,b). In short, the coefficient $\alpha$ in the linear relation between the specific attenuation at H polarization ($A_h$) and specific differential phase ($K_{dp}$) is determined by minimizing a cost function based on least squares [we refer to *Bringi and Chandrasekar* 2001 for details], whereas the standard ZPHI method uses a fixed *a priori* value for $\alpha$ (*Testud et al.* 2000). In addition, a power law of the form $A_h = b_2 Z_h^{b_1}$ is assumed where $b_1 = 0.78$ and $b_2$ are constants (*Park et al.* 2005a). The method gives the estimate of $A_h$ at each resolution volume in the selected range interval (here 0-40 km). The upshot of using $A_h$ instead of $K_{dp}$ is that the former closely follows the variations in $Z_h$ without the smoothing needed for estimating the latter, but with all the advantages of $K_{dp}$ such as immunity to calibration offsets and partial beam blockage *Ryzhkov et al.* 2014. There are many variants of the attenuation-correction method at X band, as elucidated, for example, by *Anagnostou et al.* (2004) and *Gorgucci and Chandrasekar* (2005). Here, the iterative filtering method of *Hubbert and Bringi* (1995) is used to separate backscatter differential phase from the propagation phase. In essence, the estimate of $A_h$ may be considered as a by-product of attenuation correction of the measured $Z_h$ using the differential propagation phase over the selected path interval as a constraint.

The correction of the measured $Z_{dr}$ for differential attenuation is based on an extension of the method proposed by *Smyth and Illingworth* (1998) for C-band, which is described in *Bringi and Chandrasekar* (2001) as a "combined $\Phi_{dp}$-$Z_{dr}$" constraint. The extension to X-band is described in *Park et al.* (2005b), which is used herein with some modifications implemented for the CSU-CHILL radar. In brief, the $A_h$ determined by the $\Phi_{dp}$ constraint is scaled by a factor $\nu$ and the measured $Z_{dr}$ is corrected

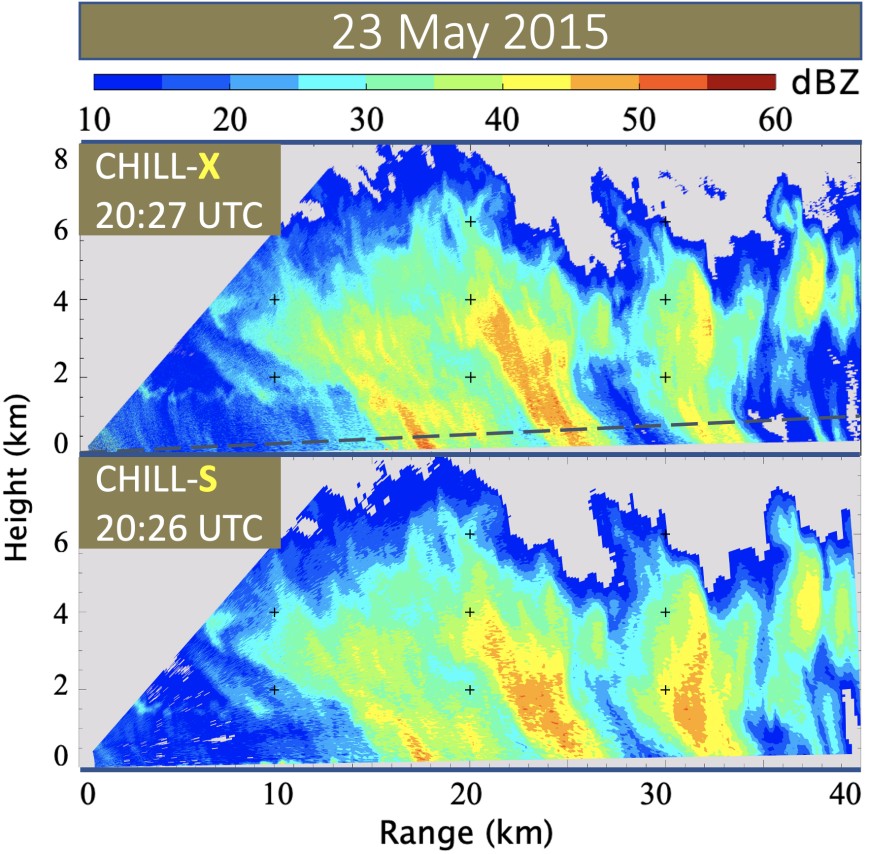

**Figure 3.** RHI sweep of $Z_H$ along the $172°$ azimuth at 2027 UTC about 18 min before peak echo descended on the Easton instrumented site at range of 13 km. The "+" marks are at 2 km intervals. (Top) X-band measured (uncorrected) reflectivity. The range profiles of radar data along the dashed line are shown in Fig. 4. (Bottom) S-band measured reflectivity.

for differential attenuation ($A_{dp} = \nu A_h$) such that a desired value is reached at the end of the beam. The desired value is the intrinsic or "true" $Z_{dr}$ at the end of the beam, which is estimated from the corrected $Z_h$ using a mean $Z_h$-$Z_{dr}$ relation based

on scattering simulations that use measured DSDs from several locations that encompass a wide variety of rain types. This sets a constraint for $Z_{dr}$ at the end of the beam (generally $Z_{dr} \approx 0$ dB because of light rain at the end of the beam or because of ice particles above the $0°$C level). By the end of the beam, we mean the last range gate where "meteo" echoes are detected. Range profiles of measured and corrected $Z_H$ and $Z_{DR}$, the measured and filtered $\Phi_{dp}$ (which is used as constraint from 0-40 km), and $A_h$ are shown in the four panels of Fig. 4 at 2027 UTC along the radial to the instrumented site located at Easton.

The $Z_H$ profiles show that very minor attenuation-correction is needed at this time, while the $Z_{DR}$ is corrected by 2 dB at the end of the ray. The change in differential phase, i.e. $\Delta\Phi_{dp}$, is also small at $25°$. Consistent with these values, the $A_h$ peak is 1.5 dB/km coinciding with the $Z_H$ peak at 25 km. At the Easton location (13 km range) the $A_h$ is negligible.

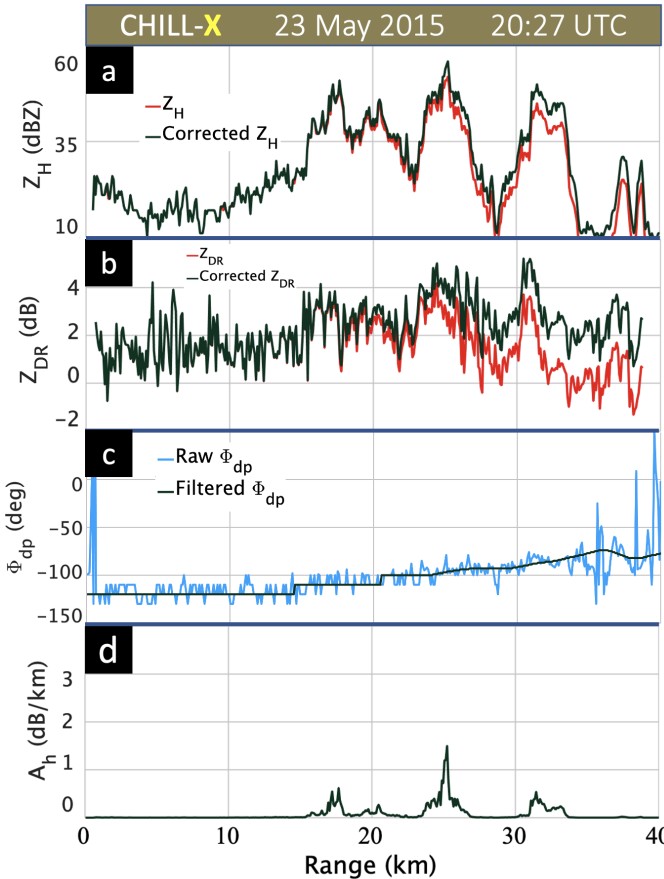

**Figure 4.** (a) Range profiles of measured and attenuation-corrected $Z_H$ at X-band at 2027 UTC at azimuth angle of $172°$ (elevation angle of $2°$). The Easton instrumented site is located at range of 13 km, (b) measured and attenuation-corrected $Z_{DR}$ at X-band, (c) measured and filtered $\Phi_{dp}$, (d) specific attenuation ($A_h$).

Figure 5a shows the PPI (at elevation angle of $1.5°$) of the measured X-band $Z_H$ at 2043 UTC at which time the peak 55 dBZ echo traversed over the instrumented site. The X-band reflectivity in Fig. 5a can be compared with S-band data in Fig. 2.
The line of cells organized south of the radar causes significant attenuation of the X-band signal power. This is clear in the range profile in Figure 5b where the attenuation has increased dramatically with $Z_H$ corrected by 35 dB and $Z_{DR}$ by 9 dB. The $\Delta\Phi_{dp}$ now increases by around $150°$. Assuming a nominal $\alpha$ of $0.25°$ km$^{-1}$, the path integrated attenuation would be 37.5 dB. The $A_h$ values have increased with peaks of 3 dB/km. At the Easton site the $A_h \approx 1.5$ dB/km. From a moment-retrieval viewpoint, significant attenuation-correction only begins beyond the Easton site (13 km range) so that the errors due to such
correction will not be significant in this case. This situation persists after 2043 UTC until the end of the analysis period (2135 UTC or so).



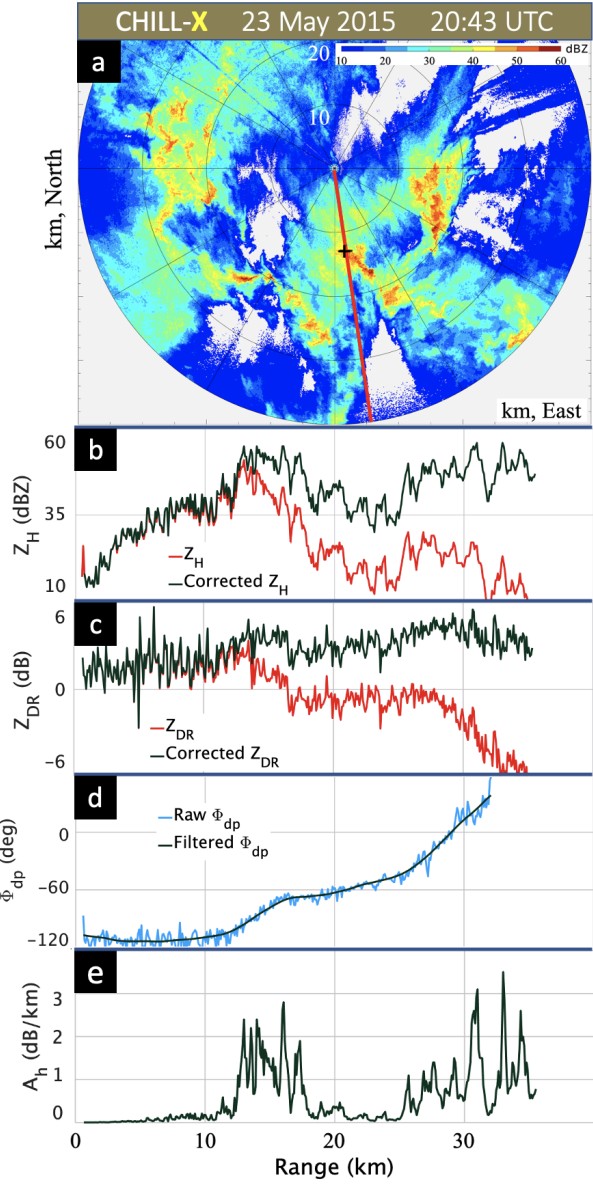

**Figure 5.** (a) PPI of measured X-band reflectivity at 2043 UTC (elevation angle is $1.5°$). The range profiles in the panels below are along the red line (radial) to the instrumented site noted by the "+" marker. (b)-(e) As in Fig. 4 panels (a)-(d), except at 2043 UTC.

## 3.3 Time Series of Radar Measurements and DSD-based Simulations

A "necessary" condition for accurate radar retrievals of DSD moments is that the time series of corrected $Z_h$, $Z_{dr}$, $K_{dp}$, and $A_h$ extracted over the resolution volumes (or, pixels) surrounding the Easton site agree "reasonably" well with the same observables
simulated using measured DSDs and a scattering model (what is generally referred to as the forward radar model/operator).





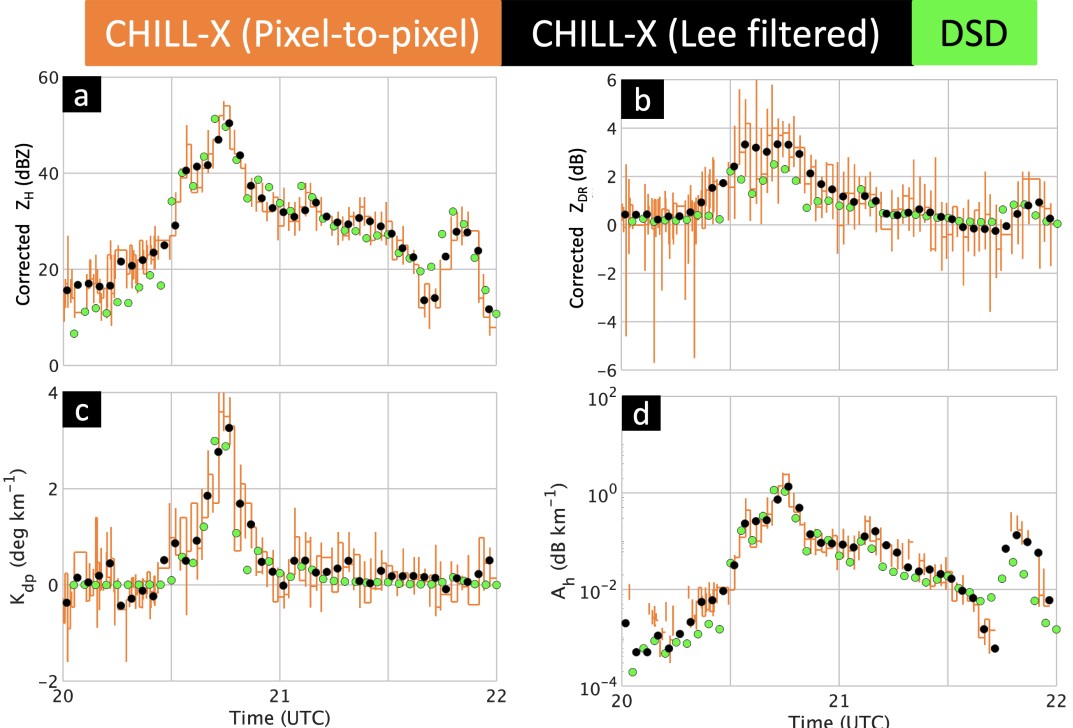

**Figure 6.** Time series of X-band radar data compared with simulations based on measured complete DSDs and scattering model described in the text. (a) Corrected $Z_H$ from radar showing pixel-to-pixel variations which have been filtered using *Lee* (1980). (b)-(d) Same but, respectively, for corrected $Z_{DR}$, $K_{dp}$ and $A_h$.

The criterion of "reasonable" agreement is difficult to quantify but elucidated in *Thurai et al.* (2012) using error variance separation. The radar data were extracted around a polar area defined by a range interval $\pm 0.18$ km centered at the range (13 km) to Easton, and $\pm 0.2°$ in azimuth angle for a total of 15 pixels surrounding the Easton site. The height of the pixels at elevation angle of $1.5°$ at 13 km range is 340 m AGL. The radar data from each pixel is plotted as a time series in Fig. 6 which

shows the pixel-to-pixel variability. A *Lee* (1980) filter (henceforth Lee filter) used to reduce speckle in images is adapted here to filter the pixel-to-pixel variability with a sliding window of $\pm 11$ (weighted) points; the filtered values are shown in Fig. 6 interpolated in time to that of the disdrometer (the radar time series were shifted by 60 s as is common to match the peak in $Z_h$). *Thurai et al.* (2012) applied the Lee filter to time series data versus range filtering applied to range gates along a fixed ray profile and showed that they were nearly equivalent. The Lee-filtered values of the radar data show the time evolution of the

main echo passage over Easton site.

The composite 3-min averaged DSDs (an example was shown in Fig. 1b) were used to simulate radar observables as a time series using the T-matrix scattering code (*Barber and Yeh* 1975; *Bringi and Seliga* 1977). The time resolution of 3-mins corresponds to spatial scale of 1.8 km (using echo movement speed of $10 \text{ m s}^{-1}$) which is less than the echo cell sizes estimated as around 2-3 km. While the DSD data were available at much higher time resolution the choice of 3-min averaged DSD is





275 a compromise between smaller DSD sample sizes when integration times are, say, 1-min, versus poorer representativeness of the spatial scales for longer time integrations (e.g., 5-min). The radar update time was around 90 s which is short enough not to introduce excessive temporal representativeness errors.

  The scattering model is based on the mean shapes from the 80-m fall bridge experiment described in *Thurai et al.* (2007) and Gaussian canting angle distribution with mean=0° and standard deviation $\sigma = 7°$ (from *Huang et al.* 2008). The dielectric
280 constant of water at wavelength of 3 cm and temperature of 8°C (Ray 1972) were used. The time series of the simulated radar observables are shown in Fig. 6 marked as "DSD". The visual agreement between simulations and the Lee-filtered mean radar values are qualitatively quite good except for a small underestimation of simulated $Z_{dr}$ relative to radar measurements by around 0.5 dB at ~2045 UTC. The discrepancy at 2140 UTC noted in Figs. 6a and d is because of heavy rain on radome (observed by PCK). Overall, the good agreement between corrected radar measurements and the DSD-based forward simulations
285 show good calibration of $Z_h$ and $Z_{dr}$. The radar-retrieved specific attenuation closely follows the $Z_h$ due to the $A_h$-$Z_h$ power law assumption in the ZPHI method (only the fixed exponent is relevant) whereas the $K_{dp}$ does not, as expected.

## 4 The Methodology of Radar Retrieval of the DSD Moments

 As mentioned in Section 1, the methodology we used here follows *RBb* except for the use of specific attenuation rather than $K_{dp}$ in the retrieval of $M_3$ (however, both methods use $Z_{dr}$ in a multi-step retrieval described below). There are several advan-
290 tages to this approach. First, "noisy" $A_h$ is strictly positive, as opposed to $K_{dp}$ which can be "noisy" with both positive and negative-valued fluctuations in measurements. This is an issue because horizontal orientation of raindrops is usually assumed for simulation of $K_{dp}$ from DSD measurements, meaning that all simulated $K_{dp}$ values used to train retrieval algorithms are positive. Second, the smoothing in range necessary for $K_{dp}$ is not needed for $A_h$ which closely follows the spatial variability in $Z_h$. The basis for the retrieval methodology lies in the double-moment normalization of *L04*. This method is explained in
295 detail in *RBb* and, hence, we only summarise it in the next subsection.

### 4.1 Overview

 Moment retrievals from polarimetric radar data are a relatively recent application of the scaling/normalization of the DSD. There are several aspects in this scaling as described by *L04* namely, there is substantial reduction in the scatter in $h(x)$ from the un-normalized scatter implying that most of the variability of the DSD can be attributed to variability in $N_0'$ and $D_m'$ with $h(x)$
300 being relatively "stable" with varying rain types/intensities. There is considerable latitude in the choice of reference moments $\{M_i, M_j\}$ in the double-moment scheme depending on the application with $N_0'$ expressed as $M_i^{(j+1)/(j-i)} M_j^{(i+1)/(i-j)}$ and $D_m'$ as ratio of $(M_j/M_i)^{1/(j-i)}$. Further, any moment $M_k$ can be expressed as power laws of $M_i$, $M_j$, and the $k^{\text{th}}$ moment of $h(x)$. *RBa* showed that the amount of variance in individual DSD moments captured by the normalization scheme depends on the choice of reference moments.

305  *RBb* first suggested the use of $\{M_3, M_6\}$ as the two reference moments suitable for polarimetric radar retrievals of the DSD. They proposed retrieval of $M_6$ from radar measurements of $Z_h$ while for $M_3$ the retrieval was based on $\{Z_{dr}, K_{dp}\}$. While





$h(x)$ can be of any functional form, the G-G model $h(x; \mu, c)$, with two positive shape parameters $\mu$ and $c$, from *L04* was chosen by *RBb*. The key to accurate retrievals of $M_k$ not only depends on the retrieval accuracy of the reference moments but also on $h(x; \mu, c)$ which has to be representative of the rain climatology. The estimation of $\{\mu, c\}$ requires a large database of DSD measurements but, more importantly, the small drop end (the fit to which is controlled mostly by $\mu$) of the distributions needs to be measured accurately as discussed in Section 2, because otherwise the lower order moments $M_0$ through $M_2$ will be in error.

In our retrieval the $h(x)$ from 1594 3-min DSDs (with rain rate $> 0.1$ mm h$^{-1}$) collected by the MPS and 2DVD in Greeley, CO (Easton) during the months from April-October 2015 formed the (Spring-Summer-Fall) "climatological" database. The $h(x)$ for each measured 3-min $N(D)$ was calculated by normalizing using $N_0'$ and scaled using $D_m'$. The median values of $h(x)$ in each bin of width $\delta x = 0.05$ were obtained and fitted to the G-G model through a weighted least-squares minimization leading to optimized values of $\mu = -0.24$ and $c = 6.03$ (see *Raupach et al.* 2019 for details of the fitting procedure). Figure 7 shows empirical $h(x)$ values as a frequency of occurrence plot on which the optimized G-G $h(x)$ is overlaid.

Note that *Thurai and Bringi* (2018) and *Raupach et al.* (2019) allowed for $\mu$ to be negative in the G-G model primarily to achieve a better fit for the small drop end of the DSD. As a result, the analytical expression for the $k^{\text{th}}$ moment $M_k$ in terms of power laws of $M_3$ and $M_6$ with $\{\mu, c\}$ cannot be used. Instead, the radar estimates of moments of the retrieved function are calculated directly from moments of fitted $N(D) = N_0' h(x)$, where $h(\cdot)$ is (*L04*)

$$h_{(i,j,\mu,c)}(x) = c\Gamma_i^{\frac{(j+c\mu)}{(i-j)}} \Gamma_j^{\frac{(-i-c\mu)}{(i-j)}} x^{c\mu-1} \exp\left[-\left(\frac{\Gamma_i}{\Gamma_j}\right)^{\frac{c}{(i-j)}} x^c\right],$$ (1)

where $\quad \Gamma_i = \Gamma\left(\mu + \frac{i}{c}\right)$ and $\Gamma_j = \Gamma\left(\mu + \frac{j}{c}\right)$ with $i = 3$ and $j = 6$.

## 4.2 Retrieval Algorithms

The retrieval algorithms for the reference moments $\{M_3, M_6\}$ are based on 2928 3-min averaged complete DSDs from GXY and HSV. The combined DSDs from both locations are used because the frequency of occurrence of significant values of $A_h$ ($> 1$ dB km$^{-1}$) from GXY alone was not enough to get a good retrieval. The scattering model assumptions are as given earlier in Section 3.3. For retrieval of $M_6$ (in mm$^6$ m$^{-3}$), the obvious choice is $Z_h$ and a power law fit was derived for three ranges of $Z_H$:

$$M_6 = 0.98 Z_h^{1.006}, \ \ Z_H < 30 \ \text{dBZ},$$ (2a)

$$M_6 = 2.19 Z_h^{0.89}, \ \ 30 \leq Z_H < 45 \ \text{dBZ},$$ (2b)

$$M_6 = 5.57 Z_h^{0.82}, \ \ Z_H \geq 45 \ \text{dBZ}.$$ (2c)

The above ranges of $Z_H$ were based on trial and error to minimize the parameterization errors (Fig. 8a). The slight decrease in the exponent from about 1 to about 0.8 as $Z_H$ increases is because of the effect of Mie scattering at X-band. The $Z_h$ in the above fits in units of mm$^6$ m$^{-3}$ and so is $M_6$.

The retrieval of $M_3$ is based on a new multiple step procedure. First, the parameter $D_m' = (M_6/M_3)^{1/3}$ is retrieved from $Z_{DR}$ which is reasonable because $Z_{DR}$ is weighted by the axis ratio of the large drops in the distribution and $D_m$ and $D_M'$ are



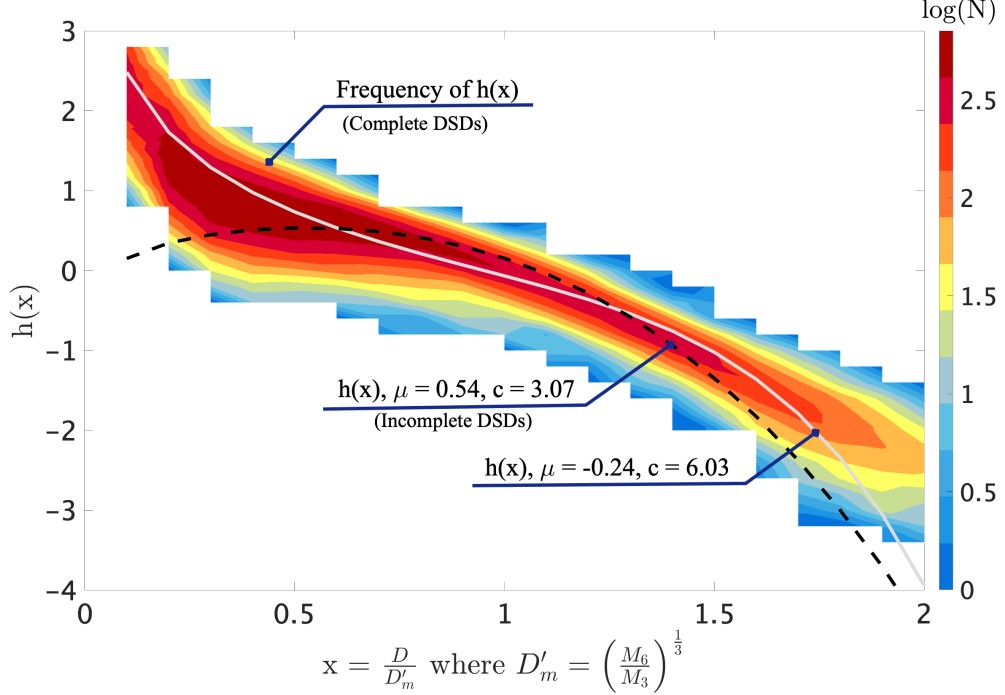

**Figure 7.** The frequency of occurrence plot of $h(x)$ from Greeley, CO with overlay of the fitted G-G ($\mu = -0.24$, $c = 6.03$). The dashed black line is $h(x)$ based on incomplete spectra using 2DVD data only. Note the y axis is on a log axis and therefore many zeros for large values of x are not shown, but still affect the per-class median values to which the fits are made.

related to the drop size. A smoothing spline fit is used as shown in Fig. 8b. Again, the intent was to reduce parameterization

errors as much as possible. The spline yields a visibly excellent fit with the $D'_m \to 0.35$ mm as $Z_{DR} \to 0$ dB. Next, the

$D_m = M_4/M_3$ is retrieved from $D'_m$ from a DSD-derived linear fit as $D_m = 0.08 + 0.8 D'_m$.

The next step is to retrieve $A_h/W$ from $D_m$ (adapted from *Jameson* 1993 who used a 3$^{\text{rd}}$ order polynomial fit) for which

we employ a smoothed spline fit. Here $A_h$ is in dB km$^{-1}$ and $W$ is the rain water content in gm$^{-3}$. We restricted the range of

$A_h/W$ at X-band to between 0.02 and 2 to avoid outliers. The smoothing spline fit is shown in Fig. 8c which again provides a

visibly good fit and is robust if the $D_m$ falls outside the specified range. The retrieval of $M_3$ follows from

$$M_3 = \frac{6000}{\pi} W = \frac{6000}{\pi} \frac{A_h}{f(D_m)}, \tag{3}$$

where $f(D_m)$ is the spline fit shown in Fig. 8c. The scatter plot of retrieved $M_3$ versus "true" $M_3$ is shown in Fig. 8d. This

multi-step procedure was devised to minimize the parameterization (or, algorithm) errors but we note it is by no means the

only way to achieve this.

It is known that the absorption cross section (specifically for X-band used here) depends on the temperature $T$ via the

$\text{Im}\{\varepsilon_r\}$, where $\varepsilon_r$ is the dielectric constant of water. For a given $W$, the integral of the extinction cross section weighted by

$N(D)$ or $A_h$ increases with colder water temperature, but also depends on $D_m$ (*Jameson* 1993). Scattering simulations were

**Figure 8.** (a) Retrieval of $M_6$ as a power law of $Z_h$ as per (2); each data point is based on a 3-min averaged complete spectra from either Greeley, CO or Huntsville, AL sites. The simulations of X-band $Z_h$, $Z_{DR}$ and $A_h$ are based on assumptions in Section 3.3. (b) Retrieval of $D'_m$ from $Z_{DR}$ along with smoothed cubic spline fit. (c) same but for retrieval of $A_h/W$ from $D_m$ where $W$ is the rain water content. (d) The retrieved $M_3$ versus "true" $M_3$. (e) Box plots of relative bias for retrieval of $M_3$ and $M_6$ (which is a measure of the deviation of the fitted values from the "true" values because of DSD variability). The inter-quartile range is given by the "height" of the dark blue box while the red horizontal line inside the blue box is the median. The outliers (orange circles) are shown in red; <10% are estimated to be outliers.

performed for 8°C and 20°C and the spline fits of $A_h/W$ versus $D_m$ were compared. For low values of $D_m$, the maximum difference was 35% occurring at $D_m = 0.75$ mm (with $A_h/W$ larger at 8° relative to 20°C as expected) but a cross-over occurs near $D_m = 1.8$ mm and the deviations increase in the opposite direction, with maximum deviation of -15% at $D_m = 3$ mm ($A_h/W$ at 20°C larger than at 8°C due to scattering loss). Recall that the National Weather Service (NWS) sounding at





Denver about 65 km away showed surface $T$ of $12°C$. A lower temperature of $8°C$ was used in the scattering calculations to approximately account for cooling of the atmosphere near Easton due to rainfall. The other temperature dependence is the coefficient $\alpha$ in the relation $A_h = \alpha K_{dp}$ used in the iterative ZPHI method. This method involves finding an optimized $\alpha$

for each beam and is assumed to account for temperature changes. Since the actual drop temperature is not known and the surface $T$ of $12°C$ is close to the assumed $T$ of $8°C$, the spline fit shown in Fig. 8c is considered to be sufficiently accurate for the retrieval of $M_3$. We note that *Diederich et al.* (2015) found that the $R(A_h)$ relation at X-band had a relatively "weak" dependence on temperature. Their fitted power law was $45.5A_h^{0.83}$ at $10°C$ to $43.5A_h^{0.79}$ at $20°C$. At $R = 10$ mm h$^{-1}$, the $A_h$ at $10°C$ is larger than at $20°C$ by $6.8\%$ while, at $100$ mm h$^{-1}$, the $A_h$ at $10°C$ is lower than at $20°C$ by $-10.8\%$; this cross-over

is consistent with our calculations above.

The evaluation of the algorithm error is done by defining the absolute bias of retrieved $M$, where $M = M_3$ or $M_6$, as $\Delta = (M(\text{retrieved}) - M(\text{"true"}))$ and the relative bias $RB = 100 * \Delta/M(\text{"}true\text{"})$ as a percentage. To show the range of the relative bias and the distribution features (such as median, $25^{\text{th}}$ and $75^{\text{th}}$ percentiles) in compact form, box plots for $M_3$ and $M_6$ are shown in Fig. 8e. The $\{25^{\text{th}}, \text{median}, 75^{\text{th}}\}$ percentile values for $M_3$ and $M_6$ are $\{-3.8, 1.7, 7.2\}$ and $\{-3.8, 0.63, 6.2\}$,

respectively. Note that the median relative bias is close to $0$ and lies at the center of the box showing very low skewness. The interquartile range (IQR) is nearly the same for both $M_3$ and $M_6$. The orange line within the blue boxes, which span the IQR or the first and the third quartiles, indicates the median. The orange horizontal lines at the extremities of the plot represent the maximum and minimum of plotted quantity. The grey horizontal lines above and below the colored blue box are whiskers within which lie extreme values that are not considered outliers. The outliers (orange circles) lie beyond the first and third

quartiles by at least $1.5$ times the IQR. The number of outliers for $M_3$ are only $6.5\%$ of the total number of samples while for $M_6$ it is $9.5\%$. It is interesting to note that the mean relative bias for $M_3$ and $M_6$ are, respectively, $-1.1\%$ and $0\%$, which are close to the median values, meaning that there is low skewness in the relative error distributions.

Histograms of $\Delta/\langle M \rangle$ showed Gaussian-like shapes (not shown here). The variances of $\Delta$ normalized by $\langle M \rangle^2$ were $0.106$ and $0.606$ for $M = M_3$ and $M = M_6$, respectively; the corresponding fractional standard errors (FSE) were, respectively, $0.32$

and $0.778$. These variances are referred to as variances due to parameterization or retrieval algorithm errors which can be added to the variances of the corresponding radar measurement errors to arrive at the total error variances. It is demonstrated (in the Appendix) that the retrieval algorithm error for $M_6$ dominates the total error variance, whereas for $M_3$ the retrieval algorithm and radar measurement errors are comparable.

## 5    Validation of Radar-Retrieved Moments

The validation procedure essentially follows methods already developed for comparing radar-retrieved rain rates with disdrometers or gages (e.g., *Bringi et al.* 2011b). The mean Lee-filtered $Z_h$, $Z_{dr}$, and $A_h$ time series data (see Fig. 6) were used to retrieve time series of $\{M_3, M_6\}$. Using Eq. (1) with the climatological $h_{\text{GG}}(x; \mu, c)$ shown in Fig. 7, the other moments (0 through 2, 4 through 5, and 7) were computed.

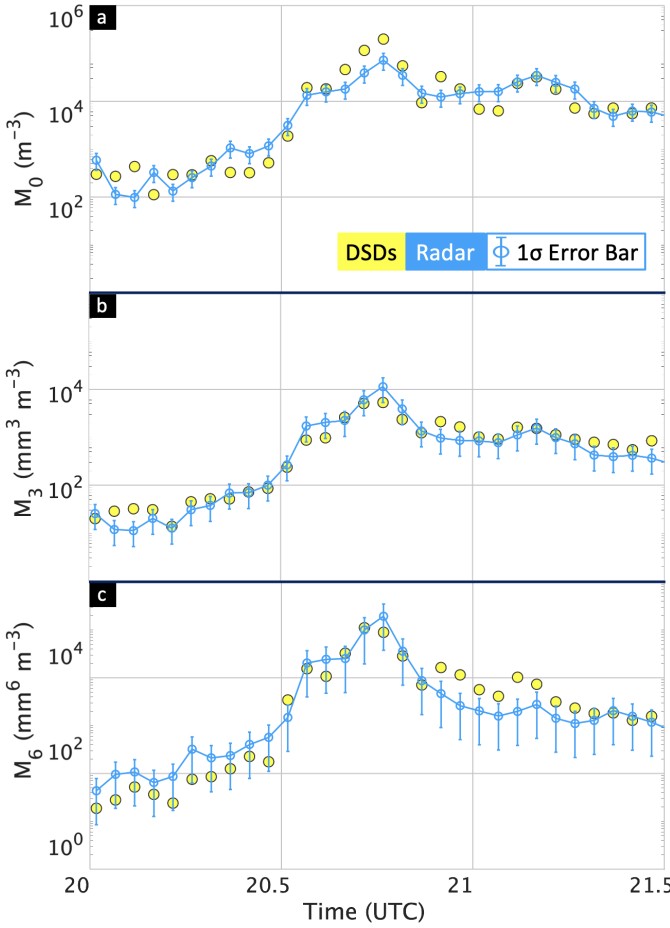

**Figure 9.** Time series of radar-derived moments and from complete DSDs over the disdrometer site. The radar estimates are mean $\pm 1\sigma$ error bars. (a) Moment $M_0$, (b)-(c) same but for $M_3$ and $M_6$.

Figure 9 shows the time series of radar-retrieved $M_0$, $M_3$, and $M_6$ with those calculated from the 3-min complete DSDs. The
radar retrievals in Fig. 9 show the mean with $\pm 1\sigma$ error bars, where $\sigma$ is the standard deviation. The mean value at each time
step is obtained from the Lee filtered values of $\{Z_h, Z_{dr}, A_h\}$ which are used to retrieve the $\{M_3, M_6\}$. Then, using Eq. (1)
the radar-retrieval of $M_0$ and other moments are obtained. The error bars or the variances consist of the sum of two terms,
namely the parameterization error variances (described above) and the radar measurement errors which are uncorrelated. The
Appendix describes the procedure to estimate the total error variances for moments such as $M_0$ and the other non-reference
moments in terms of the total error variances of $M_3$ and $M_6$. The last column in Table A1 gives the normalized total error
variances for each moment. In Fig. 9, the standard deviation is obtained at each time step by taking the square root of the
normalized variances (or, the FSE) from the last column of Table A1 for $M_0$, $M_3$, and $M_6$, with respective FSE values being





$\{0.385, 0.535, 0.805\}$. The $\sigma$ at each time step in Fig. 9 is calculated by multiplying the radar-retrieved $M_0$, $M_3$ and $M_6$ at each time step by the corresponding FSE.

Figure 9a illustrates the intercomparison of $M_0$ which is the most difficult to estimate using moments $\{M_3, M_6\}$ (see *Morrison et al.* 2019, *RBa*, and *Raupach et al.* 2019). The error bars on the radar estimates are the total errors with FSE$= 0.385$; see Appendix. The agreement with "ground truth" is visually quite remarkable considering that other error sources such as attenuation-correction or point-to-area variance, have been neglected (*Ciach and Krajewski* 1999; *Bringi et al.* 2011b). The total concentration ($M_0$) in this event ranges from 100 per $m^3$ to $100,000$ per $m^3$ at the time of peak rainfall over Easton

at 2045 UTC. Figures 9b and c show time series of $M_3$ and $M_6$, respectively. The $M_3$ retrievals are in excellent agreement with "ground truth" (total FSE$= 0.535$ with 63% of the variance due to measurement error and 37% due to algorithm error). Figure 9c compares $M_6$ and now the agreement degrades slightly. But the error bars have also increased substantially (total FSE$= 0.805$ with nearly 93% of the variance due to algorithm errors). The $M_6$ varies from 10 to $10^5$ mm$^6$ m$^{-3}$ (or equivalently 10 to 50 dBZ); the peak value at 2045 UTC being in excess of 50 dBZ. Note that $M_0$ and $M_3$ are the moments prognosed by

"bulk" double-moment numerical schemes (actually $M_0$ and mass mixing ratio). So, radar retrievals could potentially play a role in evaluating the microphysical parameterizations in such models (e.g., *Meyers et al.* 1997).

The scatter plots of $M_0$, $M_3$, and $M_6$ are shown in Fig. 10. The high correlation is quite striking and substantiated by both the Pearson's and Spearman's rank correlation values in Table 2. Note that Spearman's measures only the monotonic relationship between the correlated variables while Pearson's provides a measure of both monotonicity and linearity. The relative bias (%)

was defined in Section 4.2; for each moment ($M_0$ through $M_7$) the corresponding box plots are shown in Fig. 11. We note that there are very few or no outliers for most of the moments. Table 2 gives the median (%) and the IQR range. The IQR range is the smallest for $M_3$. This is expected because it is one of the reference moments. The median of the relative bias is "best" for $M_5$ with symmetric IQR range indicating very low skewness. The median RB for the moments $M_0$ through $M_4$ are around -15% but the skewness is significant for $M_0$ and progressively less for $M_1$ through $M_5$. The median RB for $M_6$ and $M_7$ are

$< 17\%$ but IQR indicates positive skewness (i.e., radar estimates are larger than "truth").

The difficulty in retrieving $M_0$ from higher order moments $\{M_3, M_6\}$ is clear from the box plot but nevertheless viable with relatively low median values and high correlation coefficients. However, the accuracy of all moment order retrievals, and especially the lower order, strongly depends on the climatological shape of $h(x)$ for $x < 0.75$ reflecting the shape of the small drop end (concave up for negative $\mu$). This is irrespective of well-constrained measurement and parameterization errors in the

retrieval of the reference moments $\{M_3, M_6\}$.

To illustrate this further, the "incomplete" spectra from 2DVD data alone, which are known to underestimate the numbers of small drops, are used to establish the "climatological" $h(x)$ for which the fitted G-G shape parameters are $\mu = 0.54$ and $c = 3.07$ (see Fig. 7). The radar moment retrieval steps are the same as before except for the now different $h(x)$. The "true" moments are the same as before being based solely on the complete DSD spectra. The new box plots of RB are shown in

Fig. 12. Note that now the lower order moments ($M_0$ through $M_2$) are severely underestimated with median RB slightly less than -100% but the IQR is highly compressed reflecting a distribution of RB which is concentrated as a delta function. The median RB for moments $M_6$ and $M_7$ is very large $> 400\%$ with large IQR. The moments $M_3$ through $M_5$ show more





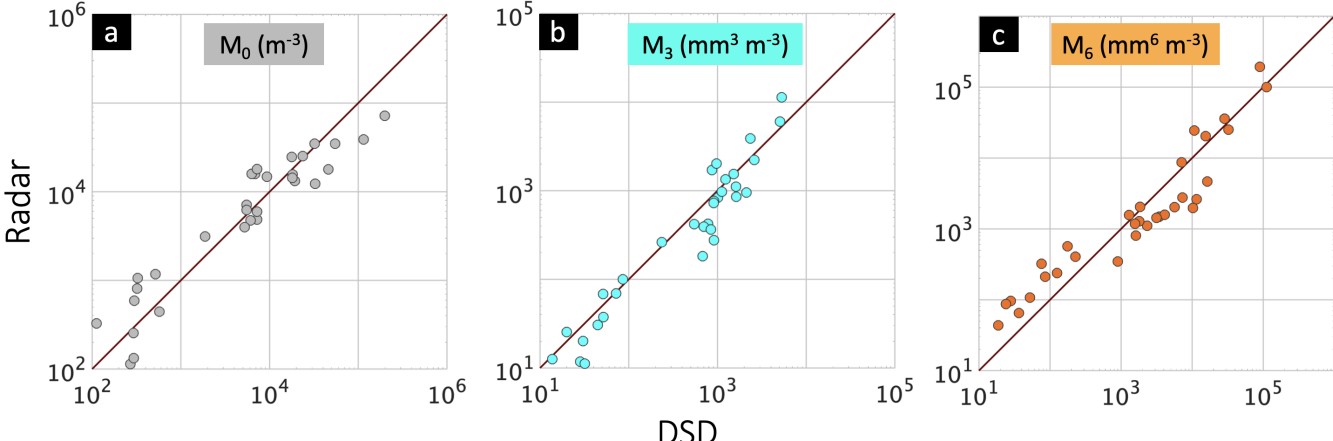

**Figure 10.** Scatter plot of radar-derived moments versus "true" moments from the complete DSD data on $\log_{10}$ scales. (a) $M_0$, (b) $M_3$ and (c) $M_6$.

**Table 2.** Statistics of the relative bias (median and IQR range). The correlations are between radar-retrieved moments and directly computed moments from the complete DSD measured by disdrometers.

| Moment | Median of RB distribution (%) | IQR ([25th, 75th] percentiles) | Spearman's rank correlation | Pearson's correlation |
|---|---|---|---|---|
| $M_0$ | -13.1 | [-34.5, 81.9] | 0.907 | 0.900 |
| $M_1$ | -17.4 | [-51, 39.5] | 0.914 | 0.924 |
| $M_2$ | -14.9 | [-47, 22.7] | 0.913 | 0.962 |
| $M_3$ | -16.5 | [-45.1, 14.1] | 0.937 | 0.906 |
| $M_4$ | -14.3 | [-39.2, 33.5] | 0.963 | 0.897 |
| $M_5$ | 4.1 | [-44.4, 61.9] | 0.973 | 0.900 |
| $M_6$ | 16.3 | [-55.7, 111.5] | 0.966 | 0.895 |
| $M_7$ | 13.6 | [-62.2, 142.5] | 0.955 | 0.881 |

"normal" RB distributions with median values of RB in the range -60 to 90%, the minimum occurring for $M_4$. However, the Pearson's correlation coefficients, as shown in Table 3, are very low for all moments implying (practically) no linear
variation between the moments. The Spearman's rank correlation is higher for moments $M_4$ through $M_7$ implying a non-linear monotonic relationship between moments probably exists. The results in Table 3 vis-à-vis Table 2 demonstrate the importance of determining the "climatological" $h(x)$ using the complete DSDs for accurate radar-based retrievals of the lower order moments. In Table 3, the floor for relative bias is $> -1$.

Finally, the radar-retrievals are examined from the perspective of identifying coherent "time tracks" as the main echo tra-
440 versed the Easton site. To this end, Fig. 13 shows tracks in, (a) the $D_m$-$M_0$ plane, (b) the $D_m$-$W$ plane, and (c) the $D'_m$-$M_6$ plane from 2012-2112 UTC. Note that we use $D'_m$ in panel (c) because it is more closely related to $Z_{dr}$. For example, panel (a)

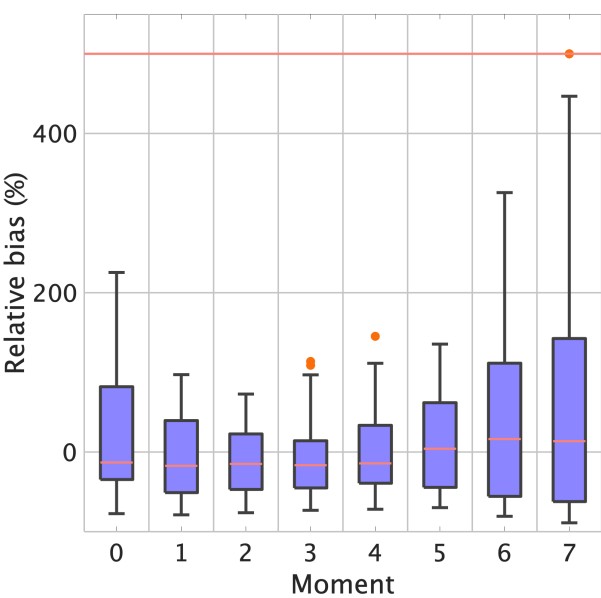

**Figure 11.** Box plots of relative bias (RB) as in Fig. 8(e) except between radar-derived moments and completed DSD moments ("truth").

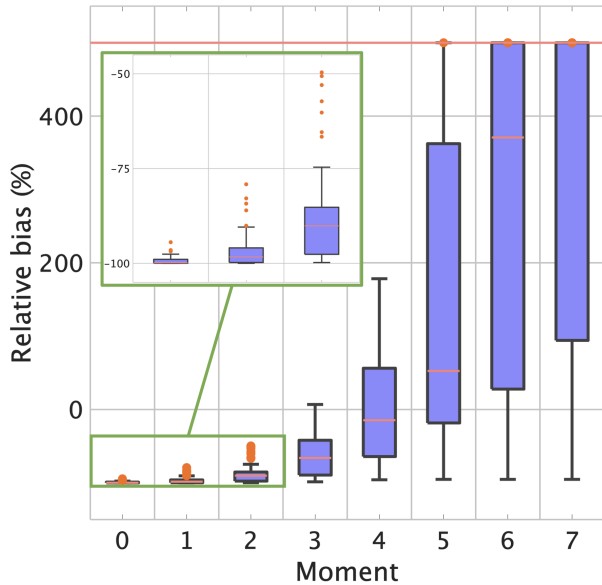

**Figure 12.** As in Fig. 11, except the incomplete 2DVD DSDs are used to determine $h(x)$ (see dashed line in Fig. 7). Inset shows magnified box plots for $M_0$ through $M_2$.

shows the initial rapid rise in $D_m$ from 1.5 mm to 2.2 mm (data point number 3 to 8 or approximately 2030-2045) with corresponding increase in $M_0$ (total number concentration) from 1000 to nearly 100,000 per m$^3$. Together with similar behavior





**Table 3.** As in Table 2, except $h(x)$ is from incomplete 2DVD DSDs only.

| Moment | Median of RB distribution (%) | IQR ([25$^{\text{th}}$, 75$^{\text{th}}$] percentiles) | Spearman's rank correlation | Pearson's correlation |
|---|---|---|---|---|
| $M_0$ | -99.869 | [-99.99, -99.12] | 0.36 | -0.025 |
| $M_1$ | -98.377 | [-99.84, -95.29] | 0.488 | -0.01 |
| $M_2$ | -89.85 | [-95.15, -79.86] | 0.65 | 0.08 |
| $M_3$ | -63 | [-84, -40.9] | 0.69 | 0.21 |
| $M_4$ | -9.5 | [-51.5, 58.4] | 0.77 | 0.284 |
| $M_5$ | 94.3 | [5.3, 384] | 0.827 | 0.27 |
| $M_6$ | 467 | [494, 1578] | 0.817 | 0.247 |
| $M_7$ | 1154 | [160, 4895] | 0.74 | 0.225 |

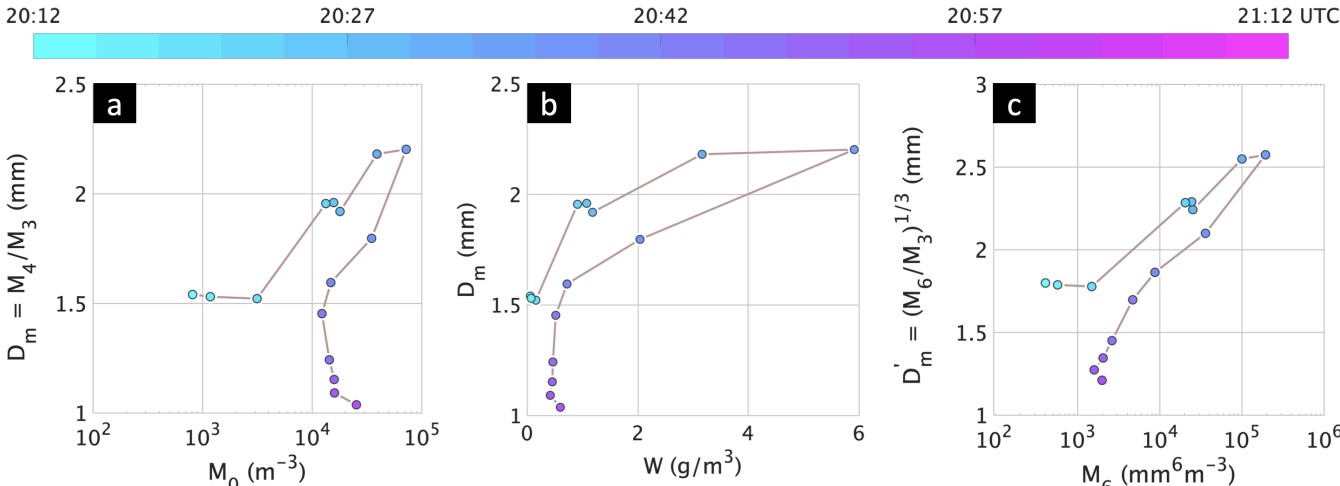

**Figure 13.** Time tracks of radar-derived variables in the (a) $D_m$ versus $M_0$ plane showing the trajectories as a function of time (color coded) over approximately an hour. Each data point reflects the radar-retrieved moments ($M_0$) or ratio of moments ($M_4/M_3$) (see Fig. 9). (b)-(c) Same but time tracks in the $D_m$ versus $W$ plane and $D'_m$ versus $M_6$ planes, respectively. Again, all quantities are from radar-retrieved moments.

in panel (b) where $W$ increases from $\leq 2$ to 6 g m$^{-3}$, and in (c) where $M_6$ increases from 30 to $> 50$ dB suggests the strong

echo aloft descending to the surface over Easton. This inference was based on examining successive volume scans from KFTG (WSR-88D in Denver, CO located about 60 km away) and noting the descent of the echo aloft to the surface at 2045 UTC. After the peak, the track (data points 8 to 11 or 2045 to 2057 UTC) reflects a rapid decrease in $M_0$ and $D_m$ and from panel (b) a rapid decrease in $W$ with corresponding rapid decrease in $M_6$, reflecting advection of the rainshaft to the north of Easton. Towards the end (last 5 data points from 2057 to 2112 UTC) the $D_m$ decrease is slowed down (from 1.5 to 1 mm) while the

$M_0$ increases modestly from $10,000$ to $15,000$ per m$^3$. At the same time the rain water content is more or less steady (at 0.5 g





m$^{-3}$) while $M_6$ decreases from 37 to 33 dB (panel b and c, last 5 data points). The echo structure during this latter time period was transitioning, from earlier descent of the strong echo over Easton, to more of a steady rain event. The decrease in $D_m$ and $M_6$ together with modest increase in $M_0$ and steady $W$ is indicative of drop-break up processes dominating the DSD evolution during this latter less convective period (*Kumjian and Prat* 2014).

**6  Discussion**

The polarimetric radar-retrieval and validation of the lower-order moments of the DSD has not received much attention in the past except for *RBb*. However, substantial literature exists in using either the un-normalized gamma model of *Ulbrich* (1983) or the normalized gamma model of *Testud et al.* (2001) to estimate the three parameters $\{N_0, \mu, \Lambda\}$ or $\{N_w, \mu, D_m\}$. As shown by *L04*, the *Testud et al.* (2001) formulation falls into double-moment normalization with reference moments $M_3$ and $M_4$ with 460  $h(x)$ being a special case of the G-G with $c = 1$, hence there is only one shape parameter $\mu$. Note that this $\mu = \mu_{\mathrm{ULB}} + 1$, where $\mu_{\mathrm{ULB}}$ is the shape parameter defined in *Ulbrich* (1983).

Many studies have attempted to retrieve the three parameters $\{N_w, \mu, D_m\}$ using polarimetric measurements $\{Z_h, Z_{dr}, K_{dp}\}$ at S-, C-, and X-bands but they are too numerous to discuss herein (e.g., *Bringi et al.* 2003; *Brandes et al.* 2003; *Park et al.* 2005b; *Gorgucci et al.* 2008; *Anagnostou et al.* 2013; to mention a few). *Anagnostou et al.* (2013) compared three different 465  methods of retrieving $N_w$ but found that validation was very difficult commenting that, "… the estimation of $N_w$ by all algorithms is significantly affected by noise or other factors like radar volume versus point (disdrometer) measurement-scale mismatch and spatial separation."

However, neither $N_0$ nor $N_w$ are the same as $M_0$ which is simply the total number concentration that scales the gamma pdf. The estimation of either $N_0$ (or $N_w$) depends on the shape parameter (or the slope parameter $\Lambda$). Typically, the $\mu$ is assumed to 470  be fixed or empirically derived as $f(\Lambda)$ or other function of $D_m$ (*Schinagl et al.* 2019). Of course, any moment of the gamma pdf can be derived as functions of the three parameters in the gamma model but very few validations of, for example, $M_0$ have been conducted. *Brandes et al.* (2003) used an empirically derived $\mu$-$\Lambda$ relation based on 2DVD data. Using $Z_h$ and $Z_{dr}$ radar measurements at S-band they retrieved $N_0$ and $\Lambda$ and obtained the $M_0$ as $N_0 \Lambda^{-(\mu+1)}$. They analyzed one convective event (with $Z_H$ varying from 10 to 55 dBZ) and showed the mean of $\log_{10}(M_0)$ from radar was 2.74 compared with 2.84 from 475  2DVD-measurements (or, 550 and 690 per m$^3$ which are much smaller than the values obtained here, see Fig.10a). The key point is that the mean $\mu$ was in the range 3-4 which is caused by truncation at the small drop end of the DSD.

*Wen et al.* (2018) describe a different method of estimating the DSD parameters of the gamma distribution and the lower order moments based on an inverse model where the input is $\{Z_{dr}, K_{dp}/Z_h\}$ and the output is $\{\mu, D_{\max}\}$ where $D_{\max}$ is the maximum diameter of the retrieved gamma DSD. Their approach follows the well-known $k$-nearest neighbour ($k$-NN) 480  classification from pattern recognition literature (*Shakhnarovich et al.* 2006). This algorithm stores all input-output associations from the available data as a "training" set. When a new $\{Z_{dr}, K_{dp}/Z_h\}$ input is presented, the algorithm assigns it the $\{\mu, D_{\max}\}$ output class that is the most common amongst the $k$ nearest (training set) neighbours of the new input. The $k$-NN is particularly suitable when large training data are available. *Wen et al.* (2018) used Euclidean distance to define the closeness of neighbours





although other distance functions are also employed in $k$-NN algorithms. They applied an empirical $\mu$-$\Lambda$ relation based on
2DVD data while $N_0$ is obtained *a posteriori* using $Z_h$, $\mu$, $\Lambda$, and $D_{\max}$. Their training set comprising $Z_{dr}$ and $K_{dp}/Z_h$ was
generated using a polynomial function whose inputs $\mu$ and $D_{\max}$ are drawn from ten-year disdrometer data with constraints
$\mu \in [-3, 20]$, $D_{\max} \in [1.7\,\mathrm{mm}, 8\,\mathrm{mm}]$, and $D_{\max} > D_m$. The test stage used S-band radar data from a WSR-88D unit (KTLX)
located in Oklahoma City, OK. A large database was analyzed and the moments $M_0$, $M_2$, $M_4$ and $M_6$ were computed from
$\{N_0, \mu, \Lambda, D_{\max}\}$. The validation results in terms of what they define as relative absolute error (RAE) ranged from 0.986 (or,
98.6%) for $M_0$ to 0.455 (or, 45.5%) for $M_6$ while the Pearson's correlation coefficient between radar-based $M_0$ and 2DVD
$M_0$ "truth" was 0.651 (the maximum correlation coefficient for other moments was $< 0.7$). The predictive performance of
$k$-NN was quantified through root relative squared error (RRSE), which computes the difference between the $k$-NN-predicted
values with the actual ones relative to when a simple predictor is used. More than characterizing the accuracy of computation
of moments, both RAE and RRSE give an indication of the efficacy of $k$-NN-based-prediction over the most basic mean-
value prediction method. *Wen et al.* (2018) reported low RRSE for $M_2$, $M_4$ and $M_6$ whereas it was large (>1) for $M_0$. They
commented that "the inverse model ... produced DSD retrievals with large uncertainties due to the measurement errors, noise,
and sampling problems of the instruments."

The *RBb* article used X-band radar $Z_h$ to retrieve $M_6$ and $\{Z_{dr}, K_{dp}\}$ to retrieve $M_3$. Our approach is similar except that we
use $A_h$ instead of $K_{dp}$. There are several advantages to using $A_h$ (similar to its use in estimating $R$ at X-band, *Diederich et al.*
(2015). For instance, $A_h$ is always positive, is highly correlated with $Z_h$ variations thus preserving the spatial resolution and,
at X-band, has decent dynamic range. The article used several networks of Parsivel disdrometer from three locations to derive
$h(x)$ and the G-G fit but their fit was more similar to that shown in Fig. 7 (black dashed line); in fact, they obtained a larger $\mu$
(2.22) and smaller $c$ (1.69). The higher value of $\mu$ gives more convex down shape at small $x$ (relative to black dashed line in
Fig. 7) while smaller $c$ results in slower fall of the tail of the distribution. The other issue they had to deal with was the "noisy"
$Z_{dr}$ and $K_{dp}$ measurements when $Z_H < 37$ dBZ. They "restored" the noisy $Z_{dr}$ by estimating it using a power law with $Z_h$
while noisy $K_{dp}$ was restored using power laws of $Z_h$ and $Z_{dr}$. They also commented that the majority of radar-measured $Z_H$
was $< 37 dBZ$. So, noise correction dominated the statistics of their moment retrievals. Their radar retrievals of moments were
based on a large dataset from three regions (two in Europe and one in Iowa, US). While they obtained median values of RB in
the range 4 to -46% their $r^2$ (squared Pearson's correlation) coefficient between radar moments and ground "truth" was very
low (0.05 to 0.33) similar to what we obtained in Table 3. They ascribed their poor correlation to spatial representativeness
errors, height of the radar pixels above the Parsivel network at longer ranges and other factors, similar to *Anagnostou et al.*
(2013).

## 7 Summary

We demonstrated a proof-of-concept of the viability of radar retrieval of lower order moments of the DSD using specific atten-
uation $A_h$ in addition to $Z_h$ and $Z_{dr}$ at X-band (an extension of *RBb*) via a case study approach. The use of specific attenuation
(from the iterative ZPHI method) is consistent with its many advantages for rain rate estimation. The multi-cell convective





complex which occurred in the area near Greeley, CO on 23 May 2015 was a target of opportunity as the CSU-CHILL radar system was available to scan the echo complex with a single elevation angle PPI every 90 s over a period of around 90 mins. The instrumented site at Easton located 13 km to the south of the radar had an MPS and 2DVD sited inside a DFIR wind shield

which made it possible to acquire the "complete" drop spectra with high resolution (50 $\mu$m) for the small drop end and good resolution (about 170 $\mu$m) for drops $\geq 0.7$ mm. The moment retrieval was based on the double-moment scaling/normalization framework of *Lee et al.* (2004). Two reference moments $\{M_3, M_6\}$ along with a "climatological" estimate of the underlying shape of the scaled/normalized DSDs fitted to the G-G distribution formed the basis of the method. The $\{M_3, M_6\}$ retrieval algorithms were trained using scattering simulations of $Z_h$, $Z_{dr}$, and $A_h$ using 2928 3-min averaged DSDs from Greeley, CO

and Huntsville, AL. The parameterization (or, algorithm) errors due to DSD variability about the smoothed spline fits were computed.

Polarimetric X-band radar data (acquired with an exceptionally narrow 3 dB beamwidth of $0.33°$) data were extracted from a small polar box surrounding the instrumented site and the moments $M_0$ through $M_7$ were estimated and validated against ground "truth" from the moments of the complete spectra using MPS and 2DVD. Using a variety of validation measures such

as box plots of relative bias, time series comparisons, scatter plots and correlation coefficients, it was determined that good accuracy was obtained for the radar-based moments well beyond than possible hitherto. For the moments $M_0$ through $M_2$, the relative bias was $< 15\%$ in magnitude with Pearson's correlation coefficients between radar-derived moments and DSD-based moments exceeding 0.9. A detailed analysis of radar fluctuations or measurement errors propagating to the variance of the moment estimates was performed; in addition, the total variance due to both parameterization and measurement errors were

tabulated.

The coherency of "time track" plots of radar retrieved quantities in the $D_m$ versus $M_0$, $D_m$ versus $W$, and $D'_m$ versus $M_6$ planes as the main 55 dBZ echo passed over the site (as well as 20 mins prior to and 20 mins after this passage) demonstrated the potential use for precipitation evolution studies for this DSD-retrieval technique. One caveat is that a much larger database is needed before concrete conclusions are drawn. In particular, the possibility of the very narrow beam of $0.33°$ and the close

range (13 km) to the instrumented site contributing to very good validation statistics, found herein relative to other studies, requires investigations with more data.

## Appendix A

The error model (*Bringi and Chandrasekar* 2001) we adopt here is an additive one, $\widehat{X} = X + \varepsilon_m + \varepsilon_p$, where $\widehat{X}$ is the estimated (or retrieved) quantity, $X$ is the "true" value, and $\varepsilon_m$ and $\varepsilon_p$ are, respectively, the radar measurement and parameterization (or

545 algorithm) errors. The $\varepsilon_m$ and $\varepsilon_p$ are zero mean, uncorrelated errors so that $\mathbb{E}\{\widehat{X}\} = X$. Thus, it follows that $\mathrm{Var}(\widehat{X} - X) = \mathrm{Var}(\varepsilon_m) + \mathrm{Var}(\varepsilon_p)$. For different rain rate estimators such as $R(Z_h)$, $R(Z_h, Z_{dr})$, and $R(K_{dp})$, the $\mathrm{Var}(\widehat{R})/\overline{R}^2$ is expressed in terms of the standard deviations of $Z_h$, $Z_{dr}$, and $K_{dp}$ which are 1 dBZ, 0.3 dB, and $0.3°$ km$^{-1}$, respectively (*Thurai et al.* 2017b).



Errors due to attenuation-correction are not considered because most of the attenuation occurred at ranges beyond the instrumented site (see Fig. 5). We refer to *Thurai et al.* (2017b) for evaluation of such errors.

## A1 Radar Measurement Errors

We consider error variances of the retrieval of $M_6$ and $M_3$ first and then the other moments. Since $M_6$ is retrieved as a power law of $Z_h^{0.8}$ (the exponent is approximate),

$$\frac{\mathrm{Var}(M_6)}{\overline{M_6}^2} = 0.8^2 \frac{\mathrm{Var}(Z_h)}{\overline{Z_h}^2}, \tag{A1}$$

where $Z_h$ has units of $\mathrm{mm^6 m^{-3}}$. Assuming the standard deviation of the radar measurement error is typically 1 dB, we get $\mathrm{Var}(Z_h)/\overline{Z_h}^2 = 0.067$. This implies

$$\mathrm{Var}\left(M_6/\overline{M_6}^2\right) = 0.043. \tag{A2}$$

The variance of $M_3$ is more complicated because it is a multiple step procedure as described in Section 4 involving smoothed spline fits. We use approximate power law fits for estimating $\mathrm{Var}(M_3)$ as follows. We have

$$D'_m \approx 1.18(Z_{dr})^{1.5}, \tag{A3}$$

and

$$\frac{A_h}{W} \approx 0.09 D_m^2, \ \text{ for } D_m > 1 \text{ mm}. \tag{A4}$$

Thus,

$$\frac{\mathrm{Var}(M_3)}{\overline{M_3}^2} = \frac{\mathrm{Var}(A_h)}{\overline{A_h}^2} + \frac{4\,\mathrm{Var}(D_m)}{\overline{D_m}^2}. \tag{A5}$$

Since $D_m$ is linear with $D'_m$, and assuming the standard deviation of the radar measurement of $Z_{DR} = 0.3$ dB,

$$\frac{\mathrm{Var}(M_3)}{\overline{M_3}^2} = \frac{\mathrm{Var}(A_h)}{\overline{A_h}^2} + \frac{9\,\mathrm{Var}(Z_{dr})}{\overline{Z_{dr}}^2}, \tag{A6}$$

where $Z_{dr}$ is a ratio, and $\mathrm{Var}(Z_{dr})/\overline{Z_{dr}}^2 = 0.0051$,

$$\frac{\mathrm{Var}(M_3)}{\overline{M_3}^2} = \frac{\mathrm{Var}(A_h)}{\overline{A_h}^2} + 0.0046. \tag{A7}$$

From *Thurai et al.* (2017b; Appendix, (A5)),

$$\frac{\mathrm{Var}(A_h)}{\overline{A_h}^2} = 0.8^2 \frac{\mathrm{Var}(Z_h)}{\overline{Z_h}^2} + \frac{\mathrm{Var}(K_{dp})}{\overline{K_{dp}}^2}, \tag{A8}$$

assuming that $A_h$ varies as $Z_h^{0.8}$ used in the ZPHI method. The standard deviation of the $K_{dp}$ measurement is typically $0.3°$ $\mathrm{km^{-1}}$ and that of $Z_H$ is 1 dB. Further, the mean $K_{dp}$ for our data $\approx 1°$ $\mathrm{km^{-1}}$ (Fig.6c) but variable in time. In any case,

$$\frac{\mathrm{Var}(A_h)}{\overline{A_h}^2} \approx (0.64)(0.067) + 0.09 = 0.133. \tag{A9}$$





Substituting above in (A6) yields

$$\frac{\mathrm{Var}(M_3)}{\overline{M_3}^2} = 0.133 + 0.046 = 0.18. \tag{A10}$$

**A2    Variances of the other Moments**

From *Lee et al.* (2004), the other moments $M_k$ can be expressed as power laws of the reference moments $M_3$ and $M_6$. They are of the form $M_k = C_k M_3^{p_k} M_6^{-q_k}$, where $p_k$ and $q_k$ are rational numbers and $C_k$ is some constant. The variance of $M_k$ needs more elaboration as $X \equiv M_3$ an $Y \equiv M_6$ are correlated. This correlation arises because $A_h$ is a power law with $Z_h$, i.e., $Z_h^{0.8}$ in the ZPHI method. Together with $M_6$ being a power law with $Z_h$, the $M_3$ and $M_6$ are correlated with a correlation coefficient of 0.93 obtained from radar-derived $M_3$ and $M_6$. For the $k$-th moment $M_k$, $p_k = \frac{6-k}{3}$ and $q_k = \frac{3-k}{3}$ for $k = 0,1,\cdots,7$, $k \neq 3,6$. The objective is to derive $\frac{\mathrm{Var}(M_k)}{\overline{M_k}^2}$ in terms of $\frac{\mathrm{Var}(M_3)}{\overline{M_3}^2}$, $\frac{\mathrm{Var}(M_6)}{\overline{M_6}^2}$, and $\mathrm{Cov}\,(M_3, M_6)$.

In the sequel, for notational simplicity, we drop the subscripts $k$. Then, for certain rational numbers $p$ and $q$, any moment $M$ is a function of these two random variables as

$$M \triangleq f\,(M_3, M_6) = C\frac{M_3^p}{M_6^q} = C\frac{X^p}{Y^q}. \tag{A11}$$

Consider the parameter vector $\boldsymbol{\theta} = (\overline{X}, \overline{Y})$. Then, second-order Taylor series approximation of $f(X,Y)$ around $\boldsymbol{\theta}$ produces

$$M \approx f(\boldsymbol{\theta}) + f_x'(\boldsymbol{\theta})(X - \overline{X}) + f_y'(\boldsymbol{\theta})(Y - \overline{Y}) + \frac{1}{2}\left\{ f_{xx}''(\boldsymbol{\theta})(X - \overline{X})^2 + 2f_{xy}''(\boldsymbol{\theta})(X - \overline{X})(Y - \overline{Y}) + f_{yy}''(\boldsymbol{\theta})(Y - \overline{Y})^2 \right\}, \tag{A12}$$

where the notations $f_x'(\boldsymbol{\theta})$ and $f_{xy}''(\boldsymbol{\theta})$ represent, respectively, the first and second-order derivatives of the function $f$ with respect to the variables in the subscript and evaluated at $\boldsymbol{\theta}$.

The second-order approximation of the mean $\overline{M} = \mathbb{E}\{M\}$ is

$$\overline{M} \approx \mathbb{E}\left\{ f(\boldsymbol{\theta}) + f_x'(\boldsymbol{\theta})(X - \overline{X}) + f_y'(\boldsymbol{\theta})(Y - \overline{Y}) + \frac{1}{2}\left\{ f_{xx}''(\boldsymbol{\theta})(X - \overline{X})^2 + 2f_{xy}''(\boldsymbol{\theta})(X - \overline{X})(Y - \overline{Y}) + f_{yy}''(\boldsymbol{\theta})(Y - \overline{Y})^2 \right\} \right\}$$

$$= \mathbb{E}\{f(\boldsymbol{\theta})\} + f_x'(\boldsymbol{\theta})\mathbb{E}\left\{(X - \overline{X})\right\} + f_y'(\boldsymbol{\theta})\mathbb{E}\left\{(Y - \overline{Y})\right\}$$

$$\quad + \frac{1}{2}\left\{ f_{xx}''(\boldsymbol{\theta})\mathbb{E}\left\{(X - X)^2\right\} + 2f_{xy}''(\boldsymbol{\theta})\mathbb{E}\left\{(X - \overline{X})(Y - \overline{Y})\right\} + f_{yy}''(\boldsymbol{\theta})\mathbb{E}\left\{(Y - \overline{Y})^2\right\} \right\}$$

$$= f(\boldsymbol{\theta}) + \frac{1}{2}\left\{ f_{xx}''(\boldsymbol{\theta})\,\mathrm{Var}(X) + 2f_{xy}''(\boldsymbol{\theta})\,\mathrm{Cov}(X,Y) + f_{yy}''(\boldsymbol{\theta})\,\mathrm{Var}(Y) \right\}, \tag{A13}$$

where the second equality results because $\mathbb{E}\{(X - \overline{X})\} = \mathbb{E}\{(Y - \overline{Y})\} = 0$. Note that, from (A13), the first-order approximation of the mean is simply $\overline{M} \approx f(\boldsymbol{\theta})$. Evaluating the function $f(\cdot)$ and its derivatives at $\boldsymbol{\theta}$ returns

$$f(\boldsymbol{\theta}) = f(\overline{X}, \overline{Y}) = C\frac{\overline{X}^p}{\overline{Y}^q}, \tag{A14}$$

$$f_{xx}''(\boldsymbol{\theta}) = Cp(p-1)\frac{\overline{X}^{p-2}}{\overline{Y}^q}, \tag{A15}$$

$$f_{xy}''(\boldsymbol{\theta}) = -Cpq\frac{\overline{X}^{p-1}}{\overline{Y}^{q+1}}, \tag{A16}$$

$$f_{yy}''(\boldsymbol{\theta}) = Cq(q+1)\frac{\overline{X}^p}{\overline{Y}^{q+2}}. \tag{A17}$$





Substituting (A14)-(A17) in (A13) yields the following approximation of the mean

$$\overline{M} \approx C\left(\frac{\overline{X}^p}{\overline{Y}^q} + \frac{p(p-1)}{2}\frac{\overline{X}^{p-2}}{\overline{Y}^q}\operatorname{Var}(X) - pq\frac{\overline{X}^{p-1}}{\overline{Y}^{q+1}}\operatorname{Cov}(X,Y) + \frac{q(q+1)}{2}\frac{\overline{X}^p}{\overline{Y}^{q+2}}\operatorname{Var}(Y)\right)$$

$$= C\frac{\overline{X}^p}{\overline{Y}^q}\left(1 + \frac{p(p-1)}{2}\frac{\operatorname{Var}(X)}{\overline{X}^2} - \frac{pq\operatorname{Cov}(X,Y)}{\overline{XY}} + \frac{q(q+1)}{2}\frac{\operatorname{Var}(Y)}{\overline{Y}^2}\right). \tag{A18}$$

In order to compute the expression of $\operatorname{Var}(M)$, we note that, by definition,

$$\operatorname{Var}(M) = \mathbb{E}\left\{(M - \overline{M})^2\right\} \approx \mathbb{E}\left\{(M - f(\boldsymbol{\theta}))^2\right\}, \tag{A19}$$

where we have used the first-order approximation of the mean $\overline{M}$. Then, ignoring all the terms above second order, the Taylor series expansion of $M$ around $\boldsymbol{\theta}$ gives

$$\operatorname{Var}(M) \approx \mathbb{E}\left\{\left(f(\boldsymbol{\theta}) + f_x'(\boldsymbol{\theta})(X - \overline{X}) + f_y'(\boldsymbol{\theta})(Y - \overline{Y}) - f(\boldsymbol{\theta})\right)^2\right\}$$

$$= \mathbb{E}\left\{\left(f_x'(\boldsymbol{\theta})(X - \overline{X}) + f_y'(\boldsymbol{\theta})(Y - \overline{Y})\right)^2\right\}$$

$$= \mathbb{E}\left\{f_x'^2(\boldsymbol{\theta})(X - \overline{X})^2 + 2f_x'(\boldsymbol{\theta})f_y'(\boldsymbol{\theta})(X - \overline{X})(Y - \overline{Y}) + f_y'^2(\boldsymbol{\theta})(Y - \overline{Y})^2\right\}$$

$$= f_x'^2(\boldsymbol{\theta})\operatorname{Var}(X) + 2f_x'(\boldsymbol{\theta})f_y'(\boldsymbol{\theta})\operatorname{Cov}(X,Y) + f_y'(\boldsymbol{\theta})\operatorname{Var}(Y). \tag{A20}$$

Again, evaluating at $\boldsymbol{\theta}$, we obtain

$$f_x'^2(\boldsymbol{\theta}) = C^2 p^2 \frac{\overline{X}^{2(p-1)}}{\overline{Y}^{2q}} \tag{A21}$$

$$f_x'(\boldsymbol{\theta})f_y'(\boldsymbol{\theta}) = -C^2 pq \frac{\overline{X}^{2p-1}}{\overline{Y}^{2q+1}}, \tag{A22}$$

$$f_y'^2(\boldsymbol{\theta}) = C^2 q^2 \frac{\overline{X}^{2p}}{\overline{Y}^{2(q+1)}}. \tag{A23}$$

Substituting the above in (A20) leads to the first-order approximation

$$\operatorname{Var}(M) \approx C^2\left(p^2 \frac{\overline{X}^{2(p-1)}}{\overline{Y}^{2q}}\operatorname{Var}(X) - 2pq\frac{\overline{X}^{2p-1}}{\overline{Y}^{2q+1}}\operatorname{Cov}(X,Y) + q^2\frac{\overline{X}^{2p}}{\overline{Y}^{2(q+1)}}\operatorname{Var}(Y)\right)$$

$$= C^2\frac{\overline{X}^{2p}}{\overline{Y}^{2q}}\left(\frac{p^2\operatorname{Var}(X)}{\overline{X}^2} - 2\frac{pq\operatorname{Cov}(X,Y)}{\overline{XY}} + \frac{q^2\operatorname{Var}(Y)}{\overline{Y}^2}\right). \tag{A24}$$

From (A18) and (A24), we obtain the desired ratio as

$$\frac{\operatorname{Var}(M)}{\overline{M}^2} \approx \frac{\frac{p^2\operatorname{Var}(X)}{\overline{X}^2} - 2\frac{pq\operatorname{Cov}(X,Y)}{\overline{XY}} + q^2\frac{\operatorname{Var}(Y)}{\overline{Y}^2}}{\left(1 + \frac{p(p-1)}{2}\frac{\operatorname{Var}(X)}{\overline{X}^2} - \frac{pq\operatorname{Cov}(X,Y)}{\overline{XY}} + \frac{q(q+1)}{2}\frac{\operatorname{Var}(Y)}{\overline{Y}^2}\right)^2}. \tag{A25}$$

If the correlation coefficient $\rho_{XY}$ between $X$ and $Y$ is known, then we replace $\operatorname{Cov}(X,Y)$ in (A25) to obtain

$$\frac{\operatorname{Var}(M)}{\overline{M}^2} \approx \frac{\frac{p^2\operatorname{Var}X}{\overline{X}^2} - 2pq\rho_{XY}\sqrt{\frac{\operatorname{Var}(X)}{\overline{X}^2}\frac{\operatorname{Var}(Y)}{\overline{Y}^2}} + q^2\frac{\operatorname{Var}(Y)}{\overline{Y}^2}}{\left(1 + \frac{p(p-1)}{2}\frac{\operatorname{Var}(X)}{\overline{X}^2} - pq\rho_{XY}\sqrt{\frac{\operatorname{Var}(X)}{\overline{X}^2}\frac{\operatorname{Var}(Y)}{\overline{Y}^2}} + \frac{q(q+1)}{2}\frac{\operatorname{Var}(Y)}{\overline{Y}^2}\right)^2}. \tag{A26}$$





**Table A1.** Variance estimates for the radar-based retrievals of moments $M_0$ through $M_7$. The $p$ and $q$ are the exponents of $M_k = M_3^p M_6^{-q}$. The normalized variances due to measurement errors are given in the 4$^{th}$ column while the total due to measurement and parameterization errors are given in the 5$^{th}$ column.

| Moment, $M_k$ | $p_k$ | $q_k$ | $\frac{\mathrm{Var}(M_k)}{\overline{M_k}^2}$ for $\varepsilon_m$ | Total $\frac{\mathrm{Var}(M_k)}{\overline{M_k}^2}$ for $\varepsilon_m + \varepsilon_p$ |
|---|---|---|---|---|
| $M_0$ | 2 | 1 | 0.388 | 0.148 |
| $M_1$ | 5/3 | 2/3 | 0.316 | 0.167 |
| $M_2$ | 4/3 | 1/3 | 0.245 | 0.211 |
| $M_3$ | 1 | 0 | 0.180 | 0.286 |
| $M_4$ | 2/3 | −1/3 | 0.122 | 0.389 |
| $M_5$ | 1/3 | −2/3 | 0.076 | 0.513 |
| $M_6$ | 0 | −1 | 0.043 | 0.649 |
| $M_7$ | −1/3 | −4/3 | 0.023 | 0.782 |

Using (A26), Table A1 gives the $\frac{\mathrm{Var}(M_k)}{\overline{M_k}^2}$ for radar measurement errors $\frac{\mathrm{Var}(M_3)}{\overline{M_3}^2} = 0.18$ and $\frac{\mathrm{Var}(M_6)}{\overline{M_6}^2} = 0.043$. From Section 4.2, the parameterization errors are $\frac{\mathrm{Var}(M_3)}{\overline{M_3}^2} = 0.106$ and $\frac{\mathrm{Var}(M_6)}{\overline{M_6}^2} = 0.606$. The total variances are obtained by adding these parameterization errors to the radar measurement errors, respectively, 0.18 and 0.043 to get 0.286 and 0.649. Thus, for $\widehat{M_6}$ the parameterization error dominates with 93% of the total variance, whereas for $\widehat{M_3}$ the measurement error dominates with 63% of the total. Using (A26) and total variances for $\widehat{M_3}$ and $\widehat{M_6}$ gives in Table A1 the total variances for the other moments. For $M_0$ through $M_2$, the total variance is smaller than the measurement variance because the covariance term in negative for those moments.

*Code availability.* The IDL, MATLAB and Fortran codes used in this article are available upon request from the corresponding author.

*Data availability.* The CSU-CHILL radar data are available by request from either the corresponding author or by request to http://www.chill.colostate.edu/w/Contacts. The MPS and 2DVD processed data are available upon request to the corresponding author.

*Sample availability.* Please refer to the web article at: http://www.chill.colostate.edu/w/DPWX/Modeling_observed_drop_size_distributions: _23_May_2015.

*Video supplement.* Animation of the radar PPI sweeps for the entire duration as well as the composite DSD spectra can be found in the same web article at: http://www.chill.colostate.edu/w/DPWX/Modeling_observed_drop_size_distributions:_23_May_2015.





*Author contributions.* Conceptualization: V.N.B., M.T., and T.H.R.; Methodology, investigation, and formal analysis: V.N.B., K.V.M, and M.T.; Data curation: P.C.K; Radar analysis: P.C.K., M.T., and V.N.B.; Writing — original draft preparation: V.N.B. and K.V.M.; Writing — review and editing: V.N.B., K.V.M, P.C.K., and T.H.R.; Supervision: V.N.B.

640

*Competing interests.* The authors declare no conflict of interest. The funders had no role in the design of this study; in the collection, analyses, or interpretation of its data; in the writing of this manuscript; or in the decision to publish these results.

*Acknowledgements.* M.T. and V.N.B. received funding to conduct this research from the National Science Foundation under Grant AGS-1901585. The CSU-CHILL radar was made available via a short 20 h project approved by the Scientific Director Prof. Steven Rutledge. The DFIR wind shield which housed the MPS and 2DVD was built at the Easton site near the CSU-CHILL radar under a prior NSF grant (P.I. Prof B. Notaroš). The 2DVD and Pluvio gage were graciously loaned to Colorado State University by Dr. Walter Petersen of NASA/Wallops Precipitation Research Facility. The MPS and 2DVD at the Huntsville, AL continue to be maintained by Dr. Patrick Gatlin and Mr. Matt Wingo of NASA.

645



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
