# Peer review of "Retrieval of Lower-Order Moments of the Drop Size Distribution using CSU-CHILL X-band Polarimetric Radar: A Case Study"

_Atmospheric Measurement Techniques, 2020_

## Referee Comment (RC1) · Anonymous Referee #1 · 10 Jul 2020

This manuscript opens a chance to retrieve the lower-order moments with dual-pol radar measurement. The accuracy in retrieval is remarkable and there are some rooms for microphysical interpretation of retrieved moments/parameters. The review recommends to accept this manuscript with a minor revision. See the below comments.

We believe the measured Z is relatively accurate and the moment(s) close to measurements should be retrieved most accurately. However, the results show the accuracy in M6 is not better than others. This should be elaborated more.

The reviewer recommends further studies as separate papers to explore microphysical evolution of precipitation systems after applying this retrieval technique.

[Figure]

More comments are below:

Line 6: → 0th moment of DSD, M0

Line 19∼20: the radar-retrieved characteristic diameter with M0.. More specific.

Line 52∼63: any moment Mk can be expressed as power laws of Mi, Mj , and the k-th moment of h(x) → any moment Mk can be expressed as power laws of Mi, Mj , in which the coefficient is and the k-th moment of h(x) and the two exponents are pre-determined by I and j.

lines 91∼94: Multi-step –minimize the parameterization errors???

Line 150: Schönhuber et al. (2008) → (Schönhuber et al. 2008)

Line 162∼169: Please further describe "drizzle mode', "shoulder" and "precipitation mode"

Fig 1: It is interesting to find two peaks at D=1.3mm and D=2.2mm. Any comments in terms of the equilibrium DSD?

Fig. 2: It is worthwhile to show the same image from the X-POL.

Fig. 6: Any better way to show the pixel-to-pixel data? Currently, they are quite confusing.

Lines 281∼283: Any explanation why ZDR is so different at ∼2045UTC?

Lines 320∼321: The authors need to elaborate this.

Line 338: D'M → D'm

Lines 347∼349: multi-step procedure: how does this minimize the overall errors? Please add more explanation.

Lines 407∼409: It is not intuitive. M6 is the closet moment that we can measure with the radar but the estimation accuracy is worse than other moments. Why? Further

detail explanation is required.

Lines 412~520: Same as the above comment. M3(least IQR) and M5 (unbiased) is the most accurate. It is understandable for M3. Why does the M5 have the least bias, not M6?

Fig. 11 and 12: What is the red line around 500?

Lines 452~454: Z was around 30~35dBZ in this later period. What will be the main reason of the dominant break-up process in such a moderate intensity?
* * *

---

## Author Comment (AC1) · 19 Jul 2020

**Response to Review Comments**

**"Retrieval of Lower-Order Moments of the Drop Size Distribution using CSU-CHILL X-band Polarimetric Radar: A Case Study"**

**V. Bringi, K. V. Mishra, M. Thurai, P. C. Kennedy, and T. H. Raupach**

We thank the editor and reviewers for their time and constructive comments. In the text below, we quote the reviewer's comments verbatim in bold and follow their comments with our responses in regular font and revised manuscript text in red. Additionally, we have numbered the reviewers' comments for clarity and reference purposes.

**Reviewer #1**

**R1.1a. This manuscript opens a chance to retrieve the lower-order moments with dual-pol radar measurement. The accuracy in retrieval is remarkable and there are some rooms for microphysical interpretation of retrieved moments/parameters. The review recommends to accept this manuscript with a minor revision. See the below comments.**

**R1.1b. We believe the measured $Z$ is relatively accurate and the moment(s) close to measurements should be retrieved most accurately. However, the results show the accuracy in M6 is not better than others. This should be elaborated more.**

**R1.1c. The reviewer recommends further studies as separate papers to explore microphysical evolution of precipitation systems after applying this retrieval technique.**

*Response:*
**1.1a.** We thank the reviewer for comments that will improve the paper.
**1.1b.** We address this comment regarding retrieval of M6 in our response to R1.15 and R1.16 below.
**1.1c.** As to future work, we do intend to use such retrievals for microphysical evolution studies. Thank you for this suggestion!

**R1.2. More comments are below:**
**Line 6 :$\rightarrow$ 0th moment of DSD, M0**

*Response:*
Thank you. We have fixed this in the revised manuscript.

**R1.3. Line $19 \sim 20$: the radar-retrieved characteristic diameter with M0.. More specific.**

*Response:*
We have replaced the concerned text in the revised manuscript as follows:
... the radar-retrieved mass-weighted mean diameter with $M_0$ ...

**R1.4. Line $52 \sim 63$ : any moment Mk can be expressed as power laws of Mi, Mj, and the k-th moment of $h(x) \rightarrow$ any moment Mk can be expressed as power laws of Mi, Mj in which the coefficient is and the k-th moment of $h(x)$ and the two exponents are pre-determined by I and j.**

*Response:*
Thank you for being precise. We have changed this accordingly in the revised manuscript.

**R1.5. lines $91 \sim 94$ : Multi-step -minimize the parameterization errors???**

*Response:*
We have modified this sentence as:
This multi-step procedure was found to minimize the parameterization errors (also referred as algorithm errors) in the estimation of $M_3$.

**R1.6. Line 150: Schönhuber et al. (2008)$\rightarrow$ ( Schönhuber et al. 2008)**

*Response:*
Thank you. We have fixed this in the revised manuscript.

**R1.7. Line $162 \sim 169$ : Please further describe "drizzle mode', "shoulder" and "precipitation mode"**

*Response:*
The term "drizzle mode" was used by Abel and Boutle (2012) to describe a peak in $N(D)$ that occurs when $D < 0.5$ mm. Our use of "shoulder" and "precipitation" modes are not precise but used here merely to go with Fig. 1. We have added the following text in the revised manuscript:
...defined by a peak in $N(D)$ occurring when $D < 0.5$ mm (Abel and Boutle 2012). The "shoulder" is the diameter range where the $N(D)$ either remains steady or falls off more "slowly" (generally found under equilibrium conditions (McFarquhar, 2004)). The precipitation range is used here for larger-sized drops after the "shoulder", if any. These ranges are used here only to illustrate Fig. 1.

References:
Abel, S. J. and Boutle, I. A.: An improved representation of the raindrop size distribution for single-moment microphysics schemes, Quarterly Journal of the Royal Meteorological Society, 138, 2151–2162, 2012.
McFarquhar, G. M.: A new representation of collision-induced breakup of raindrops and its implications for the shapes of raindrop size distributions, Journal of the Atmospheric Sciences, 61, 777–794, 2004.

**R1.8. Fig 1: It is interesting to find two peaks at $D = 1.3$ mm and D = 2.2 mm. Any comments in terms of the equilibrium DSD?**

*Response:*
We are not confident in commenting on the peaks based on the MPS data, given that this figure shows only an example of one 3-minute spectra. We do mention that the $N(D)$ is "equilibrium-like" and provide following three references.

References:
Low, T. B. and List, R.: Collision, coalescence and breakup of raindrops. Part II: Parameterization of fragment size distributions, Journal ofthe Atmospheric Sciences, 39, 1607–1619, 1982.
McFarquhar, G. M.: A new representation of collision-induced breakup of raindrops and its implications for the shapes of raindrop size distributions, Journal of the Atmospheric Sciences, 61, 777–794, 2004.
Straub, W., Beheng, K. D., Seifert, A., Schlottke, J., and Weigand, B.: Numerical investigation of collision-induced breakup of raindrops.Part II: Parameterizations of coalescence efficiencies and fragment size distributions, Journal of the Atmospheric Sciences, 67, 576–588, 2010.

**R1.9. Fig. 2: It is worthwhile to show the same image from the X-POL.**

*Response:*
Thank you for this suggestion. However, the corresponding X-band PPI for Fig. 2 is (understandably) quite attenuated and would distract the reader from the thrust of this Section.

**R1.10. Fig. 6: Any better way to show the pixel-to-pixel data? Currently, they are quite confusing.**

*Response:*
Thank you for this question. This is an established method that has been used earlier in many publications to compare radar measurements with surface instruments (e.g. Thurai

et al. 2012). We have explained this with clarity in lines 260-265.
References:
Thurai, M., Bringi, V. N., Carey, L. D., Gatlin, P., Schultz, E., and Petersen, W. A.: Estimating the accuracy of polarimetric radar-based retrievals of drop-size distribution parameters and rain rate: An application of error variance separation using radar-derived spatial correlations, Journal of Hydrometeorology, 13, 1066–1079, 2012.

**R1.11. Lines $281 \sim 283$ : Any explanation why ZDR is so different at $\sim 2045$ UTC?**

*Response:*
The discrepancy is quite small $\sim 0.5$ dB and occurs during the heaviest rain rates. While Fig. 5 shows that most of the attenuation occurs beyond the instrumented site, there is some attenuation prior to that which could have caused the discrepancy. In our experience the direct comparison between radar-measured $Z_{DR}$ (made aloft) and that computed from disdrometer DSDs (using forward model assumptions) is generally considered as "good" if the $\Delta Z_{DR} < 0.5$ dB.

**R1.12. Lines $320 \sim 321$ : The authors need to elaborate this.**

*Response:*
We have replaced 320-322 by the following text in the revised manuscript:
As a result, the analytical equation (42; *L04*) where $M_k$ is derivable exactly in terms of $[i, j; \ \mu, c; \ k]$ cannot be used. Instead eq. (43) of *L04*, reproduced in (1) below, is employed. The radar estimates of the moments ($M_k$, $k = 0, 7$) are obtained from the retrieved $M_3$ and $M_6$ and by numerical integration of the following function:

**R1.13. Line $338$ : D′M → D′m**

*Response:*
Thank you. We have fixed this in the revised manuscript.

**R1.14. Lines $347 \sim 349$ : multi-step procedure: how does this minimize the overall errors? Please add more explanation.**

*Response:*
By overall errors, we assume that the reviewer is referring to the sum of parameterization errors and measurement fluctuation errors? In lines 347-349, we only consider the parameterization errors and the steps are very clearly described (see Fig. 7). We do not claim

that we have minimized the parameterization errors by our method. In fact, we placed the caveat "... This multi-step procedure was devised to minimize the parameterization (or, algorithm) errors but we note it is by no means the only way to achieve this."

The Appendix has a clear explanation of the step-by-step procedure to estimate the total error variance for all the moments $M_0$-$M_7$.

**R1.15. Lines $407 \sim 409$ : It is not intuitive. M6 is the closet moment that we can measure with the radar but the estimation accuracy is worse than other moments. Why? Further detail explanation is required.**

*Response:*
We respond to this comment together with R1.16 below.

**R1.16. Lines $412 \sim 520$: Same as the above comment. M3(least IQR) and M5 (unbiased) is the most accurate. It is understandable for M3. Why does the M5 have the least bias, not M6?**

*Response:*
In theory, the $M_3$ and $M_6$ being the reference moments should have the lowest errors. While the reviewer is correct to state that $M_6$ is the closest moment to reflectivity, it is actually true only for Rayleigh scattering. We are using X-band radar where the larger-sized drops fall in the resonant regime and there were plenty of those during the passage of the 55 dBZ over the disdrometers. If one looks closely at Fig. 7a it can be noted that it is a piece-wise linear fit of $M_6$ vs $Z$ (for $Z < 37$ dBZ and $> 37$ dBZ). The slope of $M_6$ is smaller for $Z > 37$ dBZ relative to $Z < 37$ dBZ. This is due to resonant scattering. It follows that $Z_H$ goes between $M_5$ and $M_6$ at X-band. This is a possible reason why $M_5$ is superior at X-band relative to $M_6$ in terms of relative bias as well as Pearson correlation coefficient.
We have added in line 429:
It might be unexpected that the retrieval of $M_6$ being one of the reference moments is less accurate than $M_5$. One possible reason is that, at X-band, the larger drops are resonant-sized and the $Z_H$ does not vary as $M_6$ but rather closer to $M_5$ depending on the drop sizes. Fig. 7a, in fact, shows that the fit for $M_6$ has a smaller slope for $Z_H > 37$ dBZ because of resonant scattering.

**R1.17. Fig. 11 and 12: What is the red line around $500$?**

*Response:*
Thank you for this question. The orange lines in Figs. 11 and 12 indicate the *inner fence*

beyond which data samples are considered *extreme outliers*. We have added following explanation of the box plot in Section 4.2 of the revised manuscript:

The extremities of the blue boxes are called *hinges* which span the IQR or the first (lower hinge) and the third (upper hinge) quartiles. The orange line within the blue boxes indicates the median. The outliers (orange circles) lie beyond the first and third quartiles by at least 1.5 times the IQR. In particular, the 1.5 and 3 times the IQR limit above (below) the upper (lower) hinges of the boxes are called upper (lower) *inner fence* and *outer fence*, respectively. A point beyond an inner fence on either side is considered a *mild outlier* while a point beyond an outer fence is an *extreme outlier*. The largest value below the upper inner fence and the smallest value above the lower inner fence are indicated by shorter grey horizontal lines called *whiskers*, within which lie extreme values that are not considered outliers. If there are no points beyond a whisker, corresponding inner and outer fences are not plotted. Similarly, if there are no samples between the inner and outer fences, only inner fence is shown on the plot. Otherwise, the inner fence is generally omitted and only the outer fence is depicted. For example, Fig. 8e shows only outer fence lines on top and bottom. While plotting multiple box plots on the same figure, only a common fence line that is closest and outside of all boxes is shown.

We have added following in the caption of Fig. 8e:
The orange horizontal lines on top and bottom indicate the upper and lower outer fences, respectively.

Similarly, we have added the following in the captions of Figs. 11 and 12:
The orange horizontal line on top indicates the upper inner fence.

**R1.18.  Lines $452 \sim 454$ : $Z$ was around $30 \sim 35$ dBZ in this later period. What will be the main reason of the dominant break-up process in such a moderate intensity?**

*Response:*
The reviewer is correct in that the "time track" is not same as vertical profile. So, we have deleted the sentence on breakup process.

---

## Referee Comment (RC2) · Anonymous Referee #2 · 22 Jul 2020

This manuscript presents a very interesting new technique with promising results for improving the DSD retrieval from microwave remote sensing measurements. I find no fatal flaws in this study, but do have some minor comments, which are listed below.

Biggest, minor concern: There is a notion that attenuation at low rainfall intensity is not significant enough to allow retrieval of lower order moments with acceptable uncertainty (i.e., measurement error is too large due to relatively lower signals). The DSDs presented in Fig 8c are an example of this. Also, early in the case study (<2030 UTC) when the precipitation is relatively light, the M3 and M6 retrievals are not in very good agreement with that observed. The authors do allude to Kdp being too noisy,
which is well known at low rainfall intensities, but Ah is also derived from filtered psi-dp measurements. So it is not clear how Ah should be any better than Kdp.

Other, less minor comments:

Line 128...reference to Huntsville site in this context is irrelevant. Suggest re-wording this sentence to better clarify that the same disdrometer and wind shield configuration was used in both Greeley and Huntsville, but this case study is focused on an event captured in Greeley when there was coincident X-band radar data collected.

Fig 1a and references in the text would benefit from plotting exponential and gamma DSD to show comparison with G-G, especially since the text mentions exponential in lines 168-169.

Lines 201-203: "...good time resolution enabled validation..." could use some more theoretical elaboration or a citation that has results on the decorrelation of convective rain.

Lines 211-212: RHI first mentioned on line 211 and not defined until line 212.

Fig 3. The X- and S-band RHI scans are offset by 1-min. Aren't they obtained at the same time? Line 243: the term "meteo" is not widely known...is this referring to meteorological? Perhaps hydrometeor would be more appropriate since that is what is largely contributing to the backscatter at X-band.

Line 267: "...shifted by 60 sec as is common practice..." a few citations are warranted here.

Fig 6a...early during the event (<2030UTC) the reflectivity simulated from the DSDs is 3-6 dB lower than that measured by CHILL and yet there is no mention of this rather large discrepancy. This should be mentioned in lines 280-285 and a possible explanation provided.

Lines 313-318...concerning the optimized values of mu and c...how do the distributions

of mu and c for the climatological database in this study compare to those reported by Raupach et al. (2019), which used different case studies? In other words, we need more evidence showing the variability of the shape parameter c.

Fig 13: This is a great way to represent this data and a good tool to use for better understanding the microphysical processes at work. However, I have a minor suggestion...The color scale is not very discrete. So the plots would benefit from annotations of numbering the points sequentially to better match the reference to certain features described in the text.

---

## Author Comment (AC2) · 25 Jul 2020

**Response to Review Comments**

**"Retrieval of Lower-Order Moments of the Drop Size Distribution using CSU-CHILL X-band Polarimetric Radar: A Case Study"**

**V. Bringi, K. V. Mishra, M. Thurai, P. C. Kennedy, and T. H. Raupach**

We thank the editor and reviewers for their time and constructive comments. In the text below, we quote the reviewer's comments verbatim in bold and follow their comments with our responses in regular font and revised manuscript text in red. Additionally, we have numbered the reviewers' comments for clarity and reference purposes.

**Reviewer #2**

**R2.1. This manuscript presents a very interesting new technique with promising results for improving the DSD retrieval from microwave remote sensing measurements. I find no fatal flaws in this study, but do have some minor comments, which are listed below.**

*Response:*
We thank the reviewer for a positive evaluation of our manuscript and helpful comments to improve the paper.

**R2.2a. Biggest, minor concern: There is a notion that attenuation at low rainfall intensity is not significant enough to allow retrieval of lower order moments with acceptable uncertainty (i.e., measurement error is too large due to relatively lower signals).**
**R2.2b. The DSDs presented in Fig 8c are an example of this.**
**R2.2c. Also, early in the case study ($< 2030$ UTC) when the precipitation is relatively light, the M3 and M6 retrievals are not in very good agreement with that observed.**
**R2.2d. The authors do allude to KDP being too noisy, which is well known at low rainfall intensities, but Ah is also derived from filtered psi-dp measurements. So it is not clear how Ah should be any better than KDP.**

*Response:*
2.2a. The reviewer is correct that attenuation during light rainfall is not significant. However, in order to apply the ZPHI method, it is the the product of attenuation and the path that needs to be significant. For instance, long range of an S-band radar could compensate for low attenuation in a product. In fact, this is the motivation for ongoing considerations to adopt the direct use of $A_h$ to retrieve rain rate at S-band in the

WSR-88D network (*Ryzhkov and Zrnić* 2019). At a shorter X-band range of 40 km, a $\Delta\phi_{dp}$ of 5° is sufficient to apply ZPHI method (see chapter 10.4, *Ryzhkov and Zrnić* 2019).

2.2b. We refer the reviewer to the Appendix, wherein Eq. (A9) provides details of the measurement errors in $A_h$. In Fig. 8c, which is based on scattering simulations using measured DSDs, the increase in scatter at low $D_m$ is not because of "measurement error". Rather, it is due to DSD variability.

2.2c. We assume that the reviewer is referring to our Fig. 9, especially panel (c). This is traced back to Fig. 6a, where the DSD-based simulated $Z_H$ is about 5 dB lower than that measured by radar for time period 20:00-20:30 UTC. In the revised manuscript, we explain this in Section 3.3. as follows:

Note that in Fig. 6a, the DSD-based simulated $Z_H$ is about 5 dB lower than that measured by radar for time period 20:00-20:30 UTC. The measured $Z_H$ was 18-25 dBZ, implying very low rain rates ($\sim$ 0.5 mmh$^{-1}$) and low number density of drops sampled by the disdrometers. In addition, it follows from the RHI taken at 20:27 UTC in Fig. 3 that the cells are moderately slanted from NNW aloft, where generating cells might have formed at 5 km AGL to SSE at surface. Given the unsteady conditions in this complex of echoes, it is not surprising that the disdrometer-based $Z_H$ calculation is biased low by around 5 dB relative to low radar $Z_H$ values of 18-25 dBZ. These problems are mitigated when heavier rain rates traverse the site about 15 mins later.

2.2d. The reviewer raised an important concern. Note that, in the computation of $A_h$ (see Eq. (7.150), p. 505, *Bringi and Chandrasekar* 2001, itself based on *Testud et al.* 2000), the $\Psi_{dp}$ is used only as a final value constraint, i.e., its value at the end of the beam (relative to the initial system offset value). As shown in Fig. 5, the beam ends at 40 km. Moreover, it follows from Eq. (7.150) that the resolution of $A_h$ is same as that of $Z_H$ and it has the same practical advantages as that of differential phase measurements, e.g. immunity to absolute system calibration offsets in $Z_H$ and partial beam blockage, among others (see chapter 5, *Ryzhkov and Zrnić* 2019).

On the other hand, $K_{dp}$ involves the derivative of $\Psi_{dp}$ or the slope which needs to be calculated over a finite range interval. This leads to loss of resolution relative to the resolution of $A_h$ and $Z_H$. In particular, the smoothing of $K_{dp}$ is readily observed in high reflectivity compact cells with loss of resolution.

Note that some caveats do remain while using eq (7.150). First, the exponent of a $A_h$-$Z_H$ power law is fixed based on scattering simulations. Second, the coefficient $\alpha$ in the relation $A_h = \alpha K_{dp}$ is either estimated as per the procedure in chapter 10, section 4, *Ryzhkov and Zrnić* (2019), assumed fixed to its most probable value based on scattering simulations or estimated using the method given in eq (7.153), pp 506, *Bringi and Chandrasekar* (2001).

To summarize, $A_h$ is "better" than $K_{dp}$ in pure rain with compact convective cores of

high $Z_H$ for the purposes of retrieving $W$ in a multi-step procedure as described in our paper.

In the revised manuscript, we clarify this in Section 3.2 as follows:
Note that using $A_h$ for retrieval of $W$ is restricted to precipitation comprising pure rain. In contrast, using $K_{dp}$ (as in $RBb$) in pure rain entails spatial (range) smoothing which, in compact convective rain cells, "distorts" the spatial representation of the rain rate profile depicted by $Z_H$. In our multi-step retrieval procedure, it is reasonable to not mix different smoothing scales for the radar observables.

References:
Bringi, V. N. and Chandrasekar, V.: Polarimetric Doppler weather radar: Principles and applications, Cambridge University Press, 2001.
Ryzhkov, A. V. and Zrnić, D. S.: Radar polarimetry for weather observations, Springer, 2019.
Testud, J., Le Bouar, E., Obligis, E., and Ali-Mehenni, M.: The rain profiling algorithm applied to polarimetric weather radar, Journal of Atmospheric and Oceanic Technology, 17, 332–356, 2000.

**R2.3. Other, less minor comments:**
**Line 128 ... reference to Huntsville site in this context is irrelevant. Suggest re-wording this sentence to better clarify that the same disdrometer and wind shield configuration was used in both Greeley and Huntsville, but this case study is focused on an event captured in Greeley when there was coincident X-band radar data collected.**

*Response:*
Thank you for this suggestion. We have revised the text as follows.
Our retrieval algorithms (see Section 4) of the reference moments $M_3$ and $M_6$ were based on scattering simulations from the combined DSD data from two sites, namely Greeley, Colorado (GXY) and Huntsville, Alabama (HSV). The same disdrometer and wind shield configuration were deployed at both locations. However, the case study in this paper concerns the event of 23 May 2015 captured in Greeley, which also has a coincident CHILL X-band radar.

**R2.4. Fig 1a and references in the text would benefit from plotting exponential and gamma DSD to show comparison with G-G, especially since the text mentions exponential in lines $168 - 169$.**

*Response:*
Thank you for this suggestion. We have added exponential and standard gamma fits in

Fig. 1c (reproduced below) in the revised manuscript.

[Figure]

Figure 1: (a) Conceptual illustration of the complete DSD comprising the drizzle mode, the "shoulder" region and the precipitation mode. The incomplete DSD is due to drop truncation by instruments that cannot measure the concentration of small drops. (b) An example of measured 3-min averaged DSD ($R \approx 60$ mm h$^{-1}$) using collocated optical array probe with a 2DVD showing the separate measurements (note the high resolution of the MPS and the drop truncation of the 2DVD). The composite or compete spectrum is obtained by using the MPS for $D \leq 0.75$ mm and 2DVD for $D > 0.75$ mm. The dashed blue line is the G-G fit (with parameters $\mu = -0.3$, $c = 6$; see Eq. (1) for details) to the complete spectrum. Data from 23 May 2015 case study at 20:45 UTC. (c) Same data points as panel (b) but with the standard gamma (black) and exponential (red) fits.

We have also added the following line at the end of Section 2 of the revised manuscript: These features cannot be captured by either standard gamma or exponential fits, as shown in panel (c).

**R2.5. Lines 201-203: "... good time resolution enabled validation..." could use some more theoretical elaboration or a citation that has results on the decorrelation of convective rain.**

*Response:*
While a theoretical value of decorrelation time in a convective event is not precisely known, 90 s sampling time is "empirically" sufficient. In prior works such as *Thurai et al.* (2012), where a stratiform event with embedded convection was studied, the radar beam was stationary and pointed toward the disdrometer site at a range of about 13 km. The dwell time was set such that the radar data were available every 4 s over the disdrometer site. The autocorrelation of radar estimated $D_0$ from $Z_{DR}$ was computed using the 4 s samples and the $1/e$-decorrelation time of 200 s was in close agreement with the same from the 2DVD DSDs. For strong convection, the decorrelation time would be much shorter. Therefore, the current 90 s sampling by the radar in our case is certainly much better than the 5 min sampling by the WSR-88D radar scans, even if one considers that it may not have been sufficiently within the decorrelation time for convective events.

We have added following text in Section 3.3 of the revised manuscript:
For an estimate of the decorrelation time of radar-retrieved $D_0$, we refer to *Thurai et al.* (2012) which studied stratiform rain with embedded weak convection using 4 s samples; the $1/e$-folding time was around 200 s, where $e$ is the Napier's constant. For a highly convective case of our present case study, the decorrelation time would be substantially smaller but probably similar to our radar sampling of 90 s.

References:
Thurai, M., Bringi, V. N., Carey, L. D., Gatlin, P., Schultz, E., and Petersen, W. A.: Estimating the accuracy of polarimetric radar-based retrievals of drop-size distribution parameters and rain rate: An application of error variance separation using radar-derived spatial correlations, Journal of Hydrometeorology, 13, 1066–1079, 2012.

**R2.6. Lines 211-212: RHI first mentioned on line 211 and not defined until line 212.**

*Response:*
Thank you for pointing this out. We have fixed this in the revised manuscript.

**R2.7. Fig 3. The X- and S-band RHI scans are offset by 1-min. Aren't they obtained at the same time?**

*Response:*
Thank you for pointing this out. Although the archive files are written by separate data systems running on different computers for S- and X-band systems, a 1-minute difference noted by the reviewer should not exist. A closer examination of sweep times reveals the S- and X-band RHI files started at 20:26:45 and 20:26:46 UTC, respectively; this one-second difference is typical. We have updated the time for each scan to 20:27 UTC in the revised

manuscript.

**R2.8. Line 243: the term "meteo" is not widely known ... is this referring to meteorological? Perhaps hydrometeor would be more appropriate since that is what is largely contributing to the backscatter at X-band.**

*Response:*
Thank you for this suggestion. We have replaced "meteo" by hydrometeor in the revised manuscript.

**R2.9. Line 267: "... Shifted by 60 sec as is common practice..." a few citations are warranted here.**

*Response:*
Thank you for this suggestion. We refer the reviewer to May et al, (1999), which states "Obviously, there are important sampling issues inherent in the radar–gauge comparisons. The gauge data represent a time average at a particular location. For this study, the $R$ comparisons are based on $R$ derived from gauge data averaged over 5 min. These averages have been calculated centered within 1-min of the radar scan time and with delays of 2, 3, and 4-min to produce a correction for the time taken for the precipitation to reach the ground from the three radar elevations. These delays are incorporated in all the statistics and plots except where explicitly stated otherwise."

Accordingly, we have added following text in Section 3.3 of the revised manuscript:
The radar time series were shifted by 60 s as is common to match the peak in $Z_h$ (e.g., *May et al.* 1999). A more general analysis of the error characterization of radar-gauge comparison is given in *Anagnostou et al.* (1999). However, such an analysis is not needed herein because of the narrow antenna beam and short range to the instrumented site.

References:
Anagnostou, E. N., Krajewski, W. F., and Smith, J.: Uncertainty Quantification of Mean-Areal Radar-Rainfall Estimates, Journal of Atmospheric and Oceanic Technology, 16, 206-215, 1999.
May, P. T., Keenan, T. D., Zrnić, D. S., Carey, L. D., and Rutledge, S. A.: Polarimetric radar measurements of tropical rain at a 5-cm wavelength, Journal of Applied Meteorology, 38, 750-765, 1999.

**R2.10. Fig 6a ... early during the event ($< 2030$UTC) the reflectivity simulated from the DSDs is 3-6 dB lower than that measured by CHILL and yet there is no mention of this rather large discrepancy. This should be mentioned**

**in lines 280-285 and a possible explanation provided.**

*Response:*
This comment is similar to R2.2c. We refer the reviewer to our response to R2.2c.

**R2.11. Lines 313-318....concerning the optimized values of mu and c...how do the distributions of mu and c for the climatological database in this study compare to those reported by Raupach et al. (2019), which used different case studies? In other words, we need more evidence showing the variability of the shape parameter c.**

*Response:*
Thank you for raising this interesting question. For the retrieval technique presented in this paper, we are concerned with finding the best values of $c$ and $\mu$ and, therefore, do not use their distributions. Rather, we fit a generalised gamma distribution on the median values of $h(x)$ per normalised size bin. In Raupach et al. (2019), different methods for fitting such "overall" values of $c$ and $\mu$ were tested. The method we use is the one that produced the best overall performance in that previous study. A comparison with Table 3 of Raupach et al. (2019) shows that our $c$ and $\mu$ fall within the 75[th] percentile and median of 1-min $c$ and $\mu$ values in the dataset used in Raupach et al. (2019), respectively. This similarity provides confidence that our $c$ and $\mu$ values are reasonable. The validation of the technique from the previous study shows that the resulting $c$ and $\mu$ values are representative of the "overall" shape of $h(x)$.

We have added the following text in Section 4.1 of the revised manuscript:
The optimised values of $c$ and $\mu$ fall within the range of values fitted to one-minute DSDs reported by *Raupach et al.* (2019).

References:
Raupach, T. H., Thurai, M., Bringi, V. N., and Berne, A.: Reconstructing the drizzle mode of the raindrop size distribution using double-moment normalization, Journal of Applied Meteorology and Climatology, 58, 145-164, 2019.

**R2.12. Fig 13: This is a great way to represent this data and a good tool to use for better understanding the microphysical processes at work. However, I have a minor suggestion... The color scale is not very discrete. So the plots would benefit from annotations of numbering the points sequentially to better match the reference to certain features described in the text.**

*Response:*
Thank you for this suggestion. We did consider adding annotations to the plot. But the

number of points in each plot are too many and too close to provide a "clean" figure. However, the accompanying text in the manuscript explains the number of points and the sequence.